# GlucoSynth: Generating Differentially-Private Synthetic Glucose Traces

**Josephine Lamp**[1,2]    **Mark Derdzinski**[2]    **Christopher Hannemann**[2]
**Joost van der Linden**[2]    **Lu Feng**[1]    **Tianhao Wang**[1]    **David Evans**[1]
[1]University of Virginia, Charlottesville, VA, USA; [2]Dexcom, USA
jl4rj@virginia.edu; {mark.derdzinski; christopher.hannemann;
joost.vanderlinden}@dexcom.com; {lu.feng; tianhao; evans}@virginia.edu

## Abstract

We focus on the problem of generating high-quality, private synthetic glucose traces, a task generalizable to many other time series sources. Existing methods for time series data synthesis, such as those using Generative Adversarial Networks (GANs), are not able to capture the innate characteristics of glucose data and cannot provide any formal privacy guarantees without severely degrading the utility of the synthetic data. In this paper we present GlucoSynth, a novel privacy-preserving GAN framework to generate synthetic glucose traces. The core intuition behind our approach is to conserve relationships amongst motifs (glucose events) within the traces, in addition to temporal dynamics. Our framework incorporates differential privacy mechanisms to provide strong formal privacy guarantees. We provide a comprehensive evaluation on the real-world utility of the data using 1.2 million glucose traces; GlucoSynth outperforms all previous methods in its ability to generate high-quality synthetic glucose traces with strong privacy guarantees.

## 1   Introduction

The sharing of medical time series data can facilitate therapy development. As a motivating example, sharing glucose traces can contribute to the understanding of diabetes disease mechanisms and the development of artificial insulin delivery systems that improve people with diabetes' quality of life. Unsurprisingly, there are serious legal and privacy concerns (e.g., HIPAA, GDPR) with the sharing of such granular, longitudinal time series data in a medical context [1]. One solution is to generate a set of synthetic traces from the original traces. In this way, the synthetic data may be shared publicly in place of the real ones with significantly reduced privacy and legal concerns.

This paper focuses on the problem of generating high-quality, privacy-preserving synthetic glucose traces, a task which generalizes to other time series sources and application domains, including activity sequences, inpatient events, hormone traces and cyber-physical systems. Specifically, we focus on long (over 200 timesteps), bounded, univariate time series glucose traces. We assume that available data does not have any labels or extra information including features or metadata, which is quite common, especially in diabetes. Continuous Glucose Monitors (CGMs) easily and automatically send glucose measurements taken subcutaneously at fixed intervals (e.g., every 5 minutes) to data storage facilities, but tracking other sources of diabetes-related data is challenging [2]. We characterize the quality of the generated traces based on three criteria— synthetic traces should (1) conserve characteristics of the real data, i.e., glucose dynamics and control-related metrics (*fidelity*); (2) contain representation of diverse types of realistic traces, without the introduction of anomalous patterns that do not occur in real traces (*breadth*); and (3) be usable in place of the original data for real-world use cases (*utility*).

37th Conference on Neural Information Processing Systems (NeurIPS 2023).

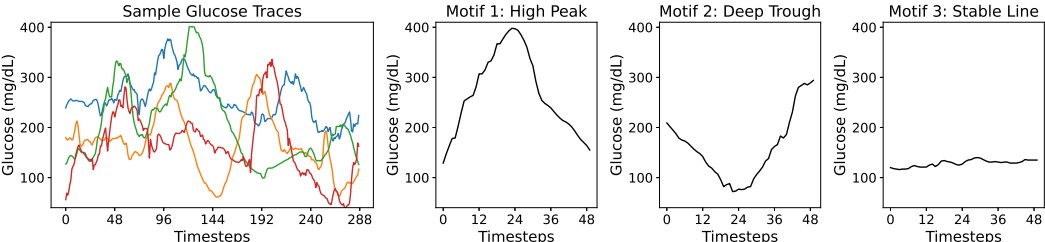

Figure 1: Example Real Glucose Traces and Glucose Motifs from our Dataset.

Generative Adversarial Networks (GANs) [3] have shown promise in the generation of time series data. However, previous methods for time series synthesis, e.g., [4, 5, 6], suffer from one or more of the following issues when applied to glucose traces: 1) surprisingly, they do not generate realistic synthetic glucose traces – in particular, they produce human physiologically impossible phenomenon in the traces; 2) they require additional information (features, metadata or labels) to guide the model learning which are not available for our traces; 3) they do not include any privacy guarantees, or, in order to uphold a strong formal privacy guarantee, severely degrade the utility of the synthetic data.

Generating high-quality synthetic glucose traces is a difficult task due to the innate characteristics of glucose data. Glucose traces can be best understood as sequences of events, which we call *motifs*, shown in Figure 1, and they are more event-driven than many other types of time series. As such, a current glucose value may be more influenced by an event that occurred in the far past compared to values from immediate previous timesteps. For example, a large meal eaten earlier in the day (30-90 minutes ago) may influence a patient's glucose more than the glucose values from the past 15 minutes. As a result, although there is some degree of temporal dependence within the traces, *only* conserving the immediate temporal relationships amongst values at previous timesteps does not adequately capture the dynamics of this type of data. In particular, we find that the main reason previous methods fail is because they may not sufficiently learn event-related characteristics of glucose traces.

**Contributions.** We present *GlucoSynth*, a privacy-preserving GAN framework to generate synthetic glucose traces. The core intuition behind our approach is to conserve relationships amongst motifs (events) within the traces, in addition to the typical temporal dynamics contained within time series. We formalize the concept of motifs and define a notion of *motif causality*, inspired from Granger causality [7], which characterizes relationships amongst sequences of motifs within time series traces (Section 4). We define a local motif loss to first train a motif causality block that learns the motif causal relationships amongst the sequences of motifs in the real traces. The block outputs a motif causality matrix, that quantifies the causal value of seeing one particular motif after some other motif. Unrealistic motif sequences (such as a peak to an immediate drop in glucose values) will have causal relationships close to 0 in the causality matrix. We build a novel GAN framework that is trained to optimize motif causality within the traces in addition to temporal dynamics and distributional characteristics of the data (Section 5). Explicitly, the generator computes a motif causality matrix from each batch of synthetic data it generates, and compares it with the real causality matrix. As such, as the generator learns to generate synthetic data that yields a realistic causal matrix (thereby identifying appropriate causal relationships from the motifs), it implicitly learns not to generate unrealistic motif sequences. We also integrate differential privacy (DP) [8] into the framework (Section 6), which provides an intuitive bound on how much information may be disclosed about any individual in the dataset, allowing the GlucoSynth model to be trained with privacy guarantees. Finally, in Section 7, we present a comprehensive evaluation using 1.2 million glucose traces from individuals with diabetes collected across 2022, showcasing the suitability of our model to outperform all previous models and generate high-quality synthetic glucose traces with strong privacy guarantees.

## 2 Related Work

We focus the scope of our comparison on current state-of-the-art methods for synthetic time series which all build upon Generative Adversarial Networks (GANs) [3] and transformation-based approaches [9]. An extended related work is in Appendix A.

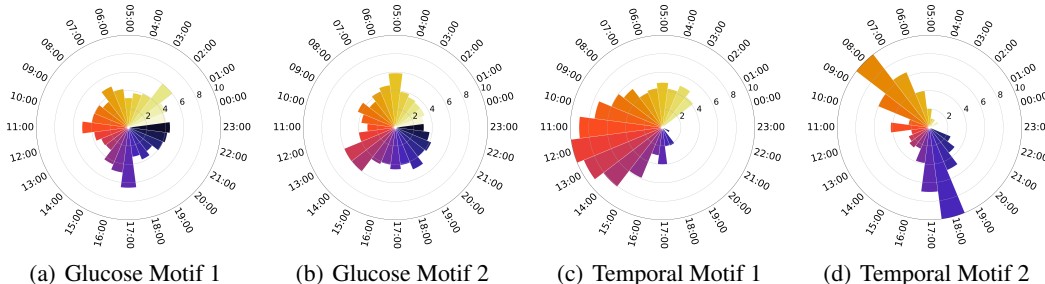

| (a) Glucose Motif 1 | (b) Glucose Motif 2 | (c) Temporal Motif 1 | (d) Temporal Motif 2 |

Figure 2: Temporal Distributions of Sample Motifs. Each radial graph displays the temporal distribution of a motif; there are 24 radial bars from 00:00 to 23:00, and each segment displays the % of motif occurrences by each hour. Glucose motifs 1 and 2 are from Fig. 1; they are not temporally-dependent and show up across the day. Temporal motifs 1 and 2 are from a cardiology dataset [15].

**Time Series.** Brophy et al. [10] provides a survey of GANs for time series synthesis. TimeGan [4] is a popular benchmark that jointly learns an embedding space using supervised and adversarial objectives in order to capture the temporal dynamics amongst traces. Esteban et al. [11] develops two time series GAN models (RGAN/RCGAN) with RNN architectures, conditioned on auxiliary information provided at each timestep during training. TTS-GAN [5] trains a GAN model that uses a transformer encoding architecture in order to best preserve temporal dynamics. Transformation-based approaches such as real-valued non-volume preserving transformations (NVP) [9] and Fourier Flows (FF) [12], have also had success for time series data. These methods model the underlying distribution of the real data to transform the input traces into a synthetic data set. Methods that only focus on learning the temporal or distributional dynamics in time series are not sufficient for generating realistic synthetic glucose traces due to the lack of temporal dependence within sequences of glucose motifs.

**Differentially-Private GANs.** To protect sensitive data, several GAN architectures have been designed to incorporate privacy-preserving noise needed to satisfy differential privacy guarantees [13]. Frigerio et al. [14] extends a simple differentially-private architecture (dpGAN) to time-series data and RDP-CGAN [6] develops a convolutional GAN architecture specifically for medical data. These methods find large gaps in performance between the non-private and private models. Providing strong theoretical DP guarantees using these methods often results in synthetic data with too little fidelity for use in real-world scenarios. Our framework carefully integrates DP into the motif causality block and each network of the GAN, resulting in a better utility-privacy tradeoff than previous methods.

## 3 Preliminaries

### 3.1 Motifs

Glucose (and many other) traces can be best understood as sequences of events or *motifs*. Motifs characterize phenomenon in the traces, such as peaks or troughs. We define a *motif*, $\mu$, as a short, ordered sequence of values ($v$) of specified length $\tau$, $\mu = [v_i, v_{i+1}, \ldots, v_{i+\tau}]$ and $\sigma$ is a tolerance value to allow approximate matching (within $\sigma$ for each value). Some examples of glucose traces and motifs are shown in Figure 1. We denote a set of $n$ time series traces as $X = [x_1, ..., x_n]$. Each time series may be represented as a sequence of motifs: $x_i = [\mu_{i_1}, \mu_{i_2}...]$ where each $i_j$ gives the index of the motif in the set that matches $x_{i_j \cdot \tau}, ... x_{i_{(j+1)} \cdot \tau - 1}$. Given the motif length $\tau$, the motif set is the union of all size-$\tau$ chunks in the traces. This definition is chosen for a straightforward implementation but motifs can be generated in other ways, such as through the use of rolling windows or signal processing techniques [16, 17]. Motifs are pulled from the data such that there is always a match from a trace motif to a motif from the set (if multiple matches, the closest one is chosen).

### 3.2 Glucose Dynamics (Why Standard Approaches Fail)

We first present a study of the characteristics of glucose data in order to motivate the development of our framework. Although there are general patterns in sequences of glucose motifs (e.g., motif patterns corresponding to patients that eat 2x vs. 3x a day), individual glucose motifs are typically not time-dependent, as illustrated in Figure 2. The radial graphs display the temporal distribution of

the first two glucose motifs from Figure 1 and two temporally-dependent motifs from a cardiology dataset [15]. There are 24 radial bars from 00:00 to 23:00 for each hour of the day, and the bar value is the percentage of total motif occurrences at that hour across the entire dataset (i.e., value of 10 would indicate that 10% of the time that motif occurs during that hour). Note that the glucose motifs show up fairly evenly *across* all hours of the day whereas the motifs from the cardiology dataset have shifts in their distribution and show up frequently at *specific* hours of the day. The lack of temporal dependence in glucose motifs is likely due to the diverse patient behaviors within a patient population. Glucose in particular is highly variable and influenced by many factors including eating, exercise, stress levels, and sleep patterns. Moreover, due to innate variability within human physiology, motif occurrences can differ even for the *same* patient across weeks or months. These findings indicate that only conserving the temporal relationships within glucose traces (as many previous methods do) may not be sufficient to properly learn glucose dynamics and output realistic synthetic traces.

### 3.3 Granger Causality

Granger causality [7] is commonly used to quantify relationships amongst time series without limiting the degree to which temporal relationships may be understood as done in other time series models, e.g., pure autoregressive ones. In this framework, an entire system (set of traces) is studied *together*, allowing for a broader characterization of their relationships, which may be advantageous, especially for long time series. We define $x_t \in \mathbb{R}^n$ as an $n$-dimensional vector of time series observed across $n$ traces and $T$ timesteps. To study causality, a vector autoregressive model (VAR) [18] may be used. A set of traces at time $t$ is represented as a linear combination of the previous $K$ lags in the series: $x_t = \sum_{k=1}^{K} A^{(k)} x_{t-k} + e_t$ where each $A^{(k)}$ is a $n \times n$ dimensional matrix that describes how lag $k$ affects the future timepoints in the series' and $e_t$ is a zero mean noise. Given this framework, we state that time series $q$ does not *Granger-cause* time series $p$, if and only if for all $k$, $A_{p,q}^{(k)} = 0$. To better represent nonlinear dynamics amongst traces, a nonlinear autoregressive model (NAR) [19], $g$, may be defined, in which $x_t = g\left(x_{1_{<t}}, ..., x_{n_{<t}}\right) + e_t$ where $x_{p_{<t}} = \left(x_{p_1}..., x_{p_{t-1}}, x_{p_t}\right)$ describes the past of series $p$. The NAR nonlinear functions are commonly modeled jointly using neural networks.

## 4 Motif Causality

Using Granger causality as defined would overwhelm the generator with too much information, resulting in convergence issues for the GAN. Instead of looking at traces comprehensively, we need a way to *scope* how the generator understands relationships between time series. To this end, we aim to use the same intuition developed from Granger causality, namely developing an understanding of relationships comprehensively using less stringent temporal constraints, but scope these relationships specifically in terms of *motifs*. Therefore, we develop a concept of *motif causality* which, by learning causal relationships amongst sequences of motifs, allows the generator to learn realistic motif sequences and produce high quality synthetic traces as a result.

### 4.1 Extending Granger Causality to Motifs

In order to quantify the relationships amongst sequences of motifs to best capture glucose dynamics, we extend the idea of Granger causality to work with motifs. Given a motif set with $m$ motifs, we build a separate (component) model, called a *motif network* in our method, for each motif, resulting in $m$ motif networks. For a single motif $\mu_i$ at time $t$, $\mu_{i_t}$, we define a function $g_i$ specifying how motifs in previous timesteps are mapped to that motif: $\mu_{i_t} = g_i\left(\mu_{1_{<t}}, ..., \mu_{m_{<t}}\right) + e_{i_t}$ where $\mu_{j_{<t}} = \left(\mu_{j_1}..., \mu_{j_{t-1}}, \mu_{j_t}\right)$ describes the past of motif $\mu_j$. The output of $g_i$ is a vector, which is added to the noise vector $e_{i_t}$. Essentially, we define motif $\mu_i$ in terms of its relationship to past motifs. The $g_i$ function takes in some *mapping* that describes how motifs in previous timesteps are mapped to the current motif $\mu_{i_t}$. The mapping is not specified in this notation, and could be defined in many different ways. In our case, we instantiate $g_i$ using a single-layer LSTM, described next.

A $g_i$ function for each motif $\mu_i$ in the motif set is modeled using a motif network with a single-layer RNN architecture. For a RNN predicting a single component motif, let $h_t \in \mathbb{R}^m$ represent the $m$-dimensional hidden state at time $t$. This represents the historical context of the motifs in the series for predicting a component motif at time $t$, $\mu_{i_t}$. At time $t$, the hidden state is updated: $h_t = g_i(h_{t-1}) + e_{i_t}$. $g_i$ here is the function describing how motifs in previous timesteps are mapped

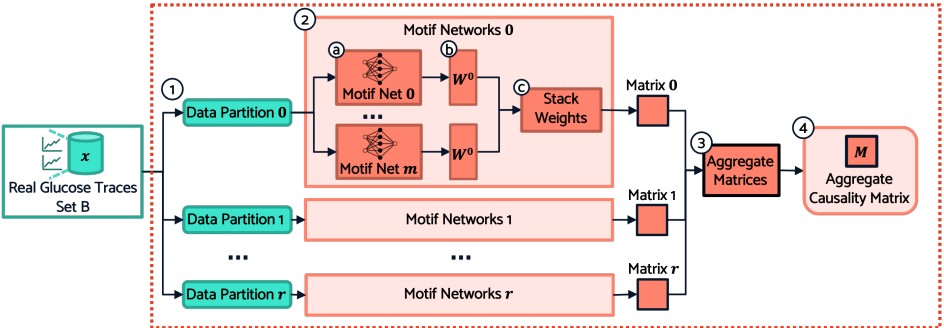

Figure 3: Motif Causality Block.

to the current motif, and is modeled (instantiated) as a single-layer LSTM as they are good at modeling long, nonlinear dependencies amongst traces [20]. The output for a motif $\mu_i$ at time $t$, $\mu_{i_t}$ can be obtained by a linear decoding of the hidden state, $\mu_{i_t} = W^o h_t + e_{i_t}$, where $W^o$ is a matrix of the output weights. These weights control the update of the hidden state and thereby control the influence of past motifs on this component motif. Essentially, this function learns a weighting that quantifies how helpful motifs in previous timesteps are for predicting the specified motif $\mu_i$ at time $t$. We note that we define causality in this way based on how Granger causality models such relationships, which is different from traditional causality models.

If all elements in the $j$th column of $W^o$ are zero ($W^o_{:j} = 0$), this is a sufficient condition for an input motif $\mu_j$ being motif non-causal on an output $\mu_i$. Therefore, we can find the motifs that are motif-causal for motif $\mu_i$ using a group lasso penalty optimization across the columns of $W^o$:

$$\min_W \sum_{t=2}^{T} (\mu_{i_t} - g_i(\mu_{0_{<t}}, ..., \mu_{m_{<t}}))^2 + \sum_{j=1}^{m} ||W^o_{:j}||_2$$

We define this as the *local motif loss*, $\mathcal{L}_{ml}$, which is optimized in each motif network using proximal gradient descent.

## 4.2 Training the Motif Causality Block

We next describe how the motif causality block is trained to learn motif causal relationships amongst traces, displayed in Figure 3. The block is structured in this way to accommodate the privacy integration (Section 6.2); here, we present its implementation without any privacy noise.

**Partition data.** First, the data is partitioned into $r$ partitions (Step 1, Figure 3) such that no models are trained on overlapping data. The number of partitions, $r$, is a user-specified hyperparameter.

**Build motif network for each motif.** Next, within each data partition a set of motif networks is trained. As a pre-processing step, we assume each trace has been chunked into a sequence of motifs of size $\tau$ (Section 3.1). $\tau$ is a hyperparameter, which we suggest chosen based on the longest effect time of a trace event. We use $\tau = 48$, corresponding to 4 hours of time, because large glucose events (from behaviors like eating) are encompassed within that time frame; see Appendix B for more details. We assume a tolerance of $\sigma = 2$ mg/dL, chosen to allow for reasonable variations in glucose. To model motif causality for an entire set of data, a $g_i$ function is implemented for each motif via a separate RNN motif net following the description provided previously, resulting in $m$ total networks (Step 2a, Figure 3). If all the motifs were trained together using a single motif network, it would not be possible to quantify the exact causal effects between each individual motif as we would not know which exact motifs contributed to a prediction (only that there is some combination of unknown motifs that contribute to an accurate prediction for a particular motif). By training each motif network separately, we are able to quantify the exact effect each motif has on each other, without any confounding effects from other motifs.

**Combine outputs of individual motif networks.** Each motif network outputs a vector of weights $W^o$ of dimensionality $1 \times m$, corresponding to the learned causal relationships (Step 2b, Figure 3). Values in the vector are between 0 (no causal relationship) and 1 (strongest causal relationship) and

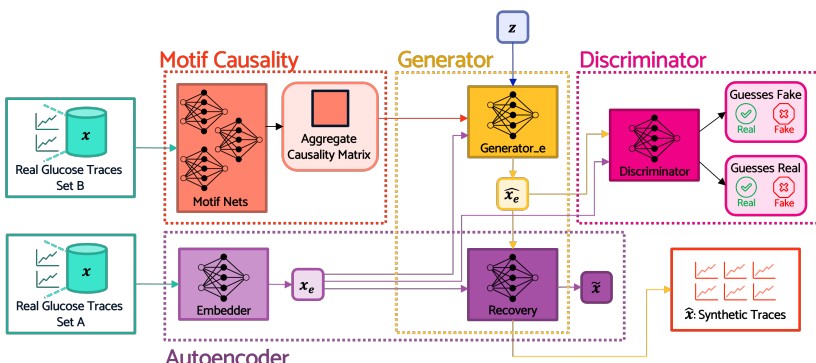

Figure 4: Overview of GlucoSynth Architecture.

give the degree to which every other motif is motif causal of the particular motif $\mu_i$ the RNN was specialized for. To return a complete matrix that summarizes causal relationships amongst *all* motifs, we stack the weights (Step 2c). The output of each data partition is a complete motif causality matrix, resulting in $r$ total matrices, each of dimensionality $m \times m$.

**Aggregate matrices and integrate with GAN.** After motif causality matrices have been outputted from each data partition, the weights in the matrices are aggregated (Step 3, Figure 3) to return the final aggregate causality matrix, $M$ (Step 4). In the nonprivate version, the weights are averaged. Finally, $M$ is sent to the generator to help it learn how to conserve motif relationships within sequences of motifs in the synthetically generated data. Details are described next in the subsequent section.

## 5 GlucoSynth

The complete GlucoSynth framework, shown in Figure 4, comprises four key blocks: the motif causality block (explained previously in Section 4), an autoencoder, a generator and a discriminator. We walk through the remaining components of the framework surrounding the GAN next.

### 5.1 GAN Architecture Components

**Autencoder.** We use an autoencoder (AE) with an RNN architecture to learn a lower dimensional representation of the traces, allowing the generator to better preserve underlying temporal dynamics of the traces. The autoencoder consists of two networks: an *embedder* and a *recovery network*. The embedder uses an encoding function to map the real data into a lower dimensional space: $Enc(x) : x \in \mathbb{R}^n \to x_e \in \mathbb{R}^e$ while the recovery network reverses this process, mapping the embedded data back to the original dimensional space: $Dec(x_e) : x_e \in \mathbb{R}^e \to \tilde{x} \in \mathbb{R}^n$. A foolproof autoencoder perfectly reconstructs the original input data, such that $x = \tilde{x} \equiv Dec(Enc(x))$. This process yields the Reconstruction Loss, $\mathcal{L}_R$, the Mean Square Error (MSE) between the original data $x$ and the recovered data, $\tilde{x}$: $\mathsf{MSE}(x, \tilde{x})$.

**Generator.** We implement the generator via an RNN or LSTM. Importantly, the generator works in the embedded space, by receiving the input traces passed through the embedder ($x_e$). To generate synthetic data, a random vector of noise, $z$ is passed through the generator and then the recovery network to return the synthetic traces in the original dimensional space. To learn how to produce high-quality synthetic data, the generator receives three key pieces of information:

*1 – Stepwise.* The generator receives batches of real data to guide the generation of realistic next step vectors. To do this, a Stepwise Loss, $\mathcal{L}_S$, is computed at time $t$ using the MSE between the batch of embedded real data, $x_{et}$, and the batch of embedded synthetic data, $\hat{x}_{et}$: $\mathsf{MSE}(x_{et}, \hat{x}_{et})$. This allows the generator to compare (and learn to correct) the discrepancies in stepwise data distributions.

*2 – Motif Causality.* The generator needs to preserve sequences of motifs in addition to temporal dynamics. Using the aggregate causality matrix $M$ returned from the Motif Causality Block, the generator computes a motif causality matrix, $M_{\hat{x}}$, on the set of synthetic data $\hat{x}$. Because the original

causality matrix was not trained on data in the embedded space, we first run the set of embedded synthetic data through the recovery network $\hat{x}_e \to \hat{x}$. From there, the Motif Causality Loss, $\mathcal{L}_M$, is computed as the MSE error between the two matrices: $\mathsf{MSE}(M, M_{\hat{x}})$. These matrices give a causal value of seeing a motif $\mu_i$ in the future after some motif $\mu_j$— unrealistic motif sequences will have causal values close to 0. As the generator learns to generate synthetic data that yields a realistic causal matrix (thereby identifying appropriate causal relationships from the motifs), it implicitly learns to not generate unrealistic motif sequences.

*3 – Distributional.* To guide the generator to produce a diverse set of traces, the generator computes a Distributional Loss, $\mathcal{L}_D$, the moments loss (MML), between the overall distribution of the real data $x_e$ and the distribution of the synthetic data $\hat{x}_e$: $\mathsf{MML}(x_e, \hat{x}_e)$. The MML is the difference in the mean and variance of two matrices.

**Discriminator.** The discriminator is a traditional discriminator model using an RNN, the only change being it also works in the embedded space. The discriminator yields the Adversarial Loss Real, $\mathcal{L}_{Ar}$, the Binary Cross Entropy (BCE) between the discriminator guesses on the real data $y_{x_e}$ and the ground truth $y$, a vector of 0's, $\mathsf{BCE}(y_{x_e}, y)$ and the Adversarial Loss Fake, $\mathcal{L}_{Af}$, the BCE between the discriminator guesses on the fake data $y_{\hat{x}_e}$ and the ground truth $y$, a vector of 1's, $\mathsf{BCE}(y_{\hat{x}_e}, y)$.

## 5.2 Training Procedure

First, the motif causality block is trained following the procedure described in Section 4.2, and then the rest of the GAN is trained. The autoencoder is optimized to minimize $\mathcal{L}_R + \alpha\mathcal{L}_S$, where $\alpha$ is a hyperparameter that balances the two loss functions. If the AE only receives $\mathcal{L}_R$ (as is typically done), it becomes overspecialized, i.e., it becomes too good at learning the best lower dimensional representation of the data such that the embedded data are no longer helpful to the generator. For this reason, the AE also receives $\mathcal{L}_S$, enabling the dual training of the generator and embedder. The generator is optimized using $\min(1 - \mathcal{L}_{Af}) + \eta(\mathcal{L}_S + \mathcal{L}_D) + \mathcal{L}_M$, where $\eta$ is a hyperparameter that balances the effect of the stepwise and distributional loss. Finally the discriminator is optimized using the traditional adversarial feedback $\min \mathcal{L}_{Af} + \mathcal{L}_{Ar}$. The networks are trained in sequence (within each epoch) in the following order: autoencoder, generator, then discriminator. In our experiments we set $\alpha = 0.1$ and $\eta = 10$ as they enable GlucoSynth to converge fastest, i.e., in the fewest epochs.

## 6   Providing Differential Privacy

There are two components to our privacy architecture, described in the following two subsections: (1) each network in the GAN (Embedder, Recovery, Generator and Discriminator networks) is trained in a differentially private manner using the Differentially-Private Stochastic Gradient Descent (DP-SGD) algorithm from Abadi et al. [21]; and (2) the motif causality block is trained using the PATE framework from Papernot et al. [22]. Importantly, two completely separate datasets are used for the training of the motif causality block (dataset B in Figure 4) and the GAN (dataset A in Figure 4). We structure the privacy integration in this way to allow for better privacy-utility trade-offs. Our design satisfies the formal differential privacy notion introduced by Dwork et al. [23]. Differential Privacy (DP) provides an intuitive bound on the amount of information that can be learned about any individual in a dataset. A randomized algorithm $\mathcal{M}$ satisfies $(\epsilon, \delta)$-differential privacy if, for all datasets $D_1$ and $D_2$ differing by at most a single unit, and all $S \subseteq \mathrm{Range}(\mathcal{M})$, $Pr[\mathcal{M}(D_1) \in S] \leq e^\epsilon Pr[\mathcal{M}(D_2) \in S] + \delta$. The parameters $\epsilon$ and $\delta$ determine the *privacy loss budget*, which provide a way to tradeoff privacy and utility; smaller values have stronger privacy. Importantly, privacy is provisioned at the *trace* level, and we assume each individual has only one trace in the dataset.

## 6.1   Training the GAN Networks with DP

To add privacy to the GAN components, each of the networks (Embedder, Recovery, Generator and Discriminator) is trained in a differentially private manner using DP-SGD [21]. Although the overall GAN framework is complicated, the individual networks all use simple RNN or LSTM architectures with Adam optimizers. As such, adding DP noise to their network weights is straightforward. We employ the following procedure using Tensorflow Privacy functions [24]. Since there are four

networks being trained with DP, we divide the privacy loss budget evenly to get the budget per network, $\epsilon_{net} = \epsilon/4$. Then, we use Tensorflow's built-in DP accountant to determine how much noise must be added to the weights of each network based on the number of epochs, batch size, number of traces and $\epsilon_{net}$. This function returns a noise multiplier, which we use when we instantiate a Tensorflow DP Keras Adam Optimizer for each network. Finally, we train each of the networks using their respective DP Keras Adam Optimizer, which automatically trains the network using DP-SGD.

## 6.2 Training the Motif Causality Block with DP

We train the motif causality block using the PATE framework [22]. PATE provides a way to return aggregated votes about the class a data point belongs to. First, the data is partitioned into $r$ partitions, where $r$ is determined based on the size of the dataset and the privacy loss budget. Then, a class membership model is trained independently for each partition. The class membership votes from each partition are aggregated by adding noise to the vote matrix and the noisiest votes are returned using the max-of-Laplacian mechanism (LNMax), tuned based on the privacy budget and $r$.

We use PATE to train the motif causality block: instead of predicting the degree of class membership we predict *causal* membership, e.g., does motif $\mu_i$ have a causal relationship to $\mu_j$. The motif causality block is trained in the same procedure described in Section 4.2 with two changes: (1) the number of data partitions, $r$, is determined based on the privacy budget, instead of a user-specified value; (2) the final causality matrix $M$ is aggregated using DP across the partitions. In normal PATE, carefully calibrated noise is added to a matrix of votes for each class, such that the classes with the noisiest votes are outputted. In our use, each value in a motif causality matrix may be likened to a class (i.e., causal "class" prediction between motif $\mu_i$ and $\mu_j$). Thus, we use the LNMax mechanism (from predefined Tensorflow Privacy functions [24]) to aggregate the matrices weights and return $M$.

We use PATE instead of training each motif network using DP-SGD for better privacy-utility trade-offs. With DP-SGD, we would need to add noise to *every* motif net, eating up our privacy budget quickly and severely impacting the quality of the returned casuality matrices. PATE allows us to train each of the motif networks without any noise on the gradients, but then aggregates their returned causality matrices in a privacy-preserving manner, resulting in a better privacy-utility trade-off.

# 7 Evaluation

Evaluating synthetic data is notoriously difficult [25], so we provide an extensive evaluation across three criteria. Synthetic data should: 1) conserve characteristics of the real data (*fidelity*, Section 7.1); 2) contain diverse patterns from the real data without the introduction of anomalous patterns (*breadth*, Section 7.2); and 3) be usable in place of the original for real-world use cases (*utility*, Section 7.3).

**Data and Benchmarks.** We use 100,000 single-day glucose traces randomly sampled across each month from January to December 2022, for a total of 1.2 million traces, collected from Dexcom's G6 Continuous Glucose Monitors (CGMs) [26]. Data was recorded every 5 minutes ($T = 288$) and each trace was aligned temporally from 00:00 to 23:59. We restrict our comparison to the five most closely related state-of-the-art models for generating synthetic univariate time series with no labels or auxiliary data: Three nonprivate—TimeGAN [4], Fourier Flows (FF) [12], non-volume preserving transformations (NVP) [9]; and two private—RGAN [11] and dpGAN [14]. We refer the reader to Appendix B for additional experimental details and all hyperparameter settings, including reasoning behind the choice of motif size $\tau = 48$.

## 7.1 Fidelity

**Visualization.** We provide visualizations of sample real and synthetic glucose traces from all models. Although this is not a comprehensive way to evaluate trace quality, it does give a snapshot view about what synthetic traces may look like. We provide heatmap visualizations, where each heatmap contains 100 randomly sampled glucose traces. Each row is a single trace from timestep 0 to 288. The values in each row indicate the glucose value (between 40 mg/dL and 400 mg/dL). Figure 5 shows the nonprivate models, and Figures 8, 9, 10 in Appendix C.1 show the private models with different privacy budgets. Upon examining the heatmaps, we notice that GlucoSynth consistently generates realistic looking glucose traces, even at very small privacy budgets.

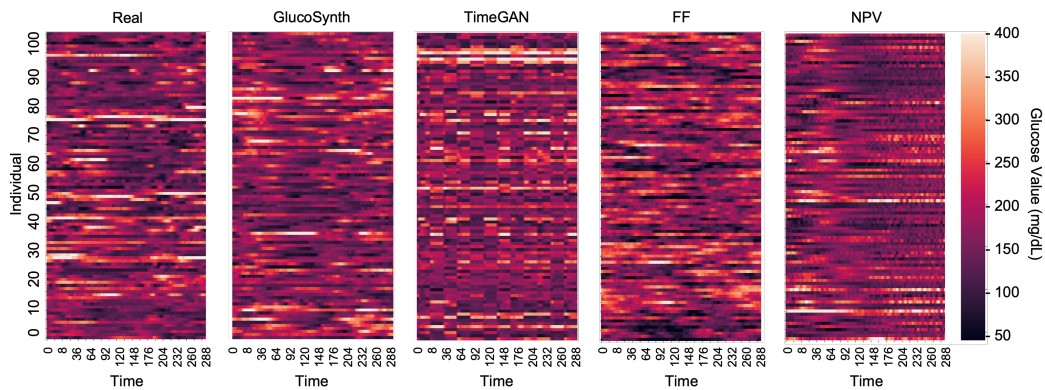

Figure 5: Heatmaps for Nonprivate Models

**Population Statistics.** To evaluate fidelity on a population scale, we compute a common set of glucose metrics and test if the difference between the synthetic and real data is statistically significant. Table 1 provides an abbreviated summary of the results; Appendix C.2 has complete results. GlucoSynth performs the best, with few statistical differences between the real and synthetic data for $\epsilon \geq 0.1$.

**Distributional Comparisons.** We visualize differences in distributions between the real and synthetic data by plotting the distribution of variances and using PCA [27]. Figure 6 shows the variance distribution for the nonprivate models. Additional comparisons across privacy budgets are available in Appendix C.3. In both nonprivate and private settings, GlucoSynth produces synthetic distributions closest to the real ones, better than all other models.

## 7.2 Breadth

We quantify breadth in terms of glucose motifs. For each model's synthetic traces, we build a motif set (see Section 3.1). Given a real motif set from the validation traces $S_x$, for each synthetic motif set $S_{\hat{x}}$, we compute "Validation Motifs", (VM), the fraction of motifs found in the validation motif set that are present in the synthetic motif set, $\text{VM}/|S_{\hat{x}}|$. This metric quantifies how good our synthetic motif set is (e.g., are its motifs mostly similar to motifs found in real traces). We also compute metrics related to *coverage*, the fraction of motifs in the validation motif set that are found in our synthetic data, defined as $\text{VM}/|S_x|$. This gives a sense of the breadth in a more traditional manner. To compare actual *distributions* of motifs (not just counts), we compute the MSE between the distribution of real motifs $S_x$ and the distribution of synthetic motifs $S_{\hat{x}}$. This gives a measure about how close the synthetic motif distribution is to the real one. We want high VM and coverage, and low MSE. Results are in Table 1 with additional analysis in Appendix D; overall our model provides the best breadth.

## 7.3 Utility

We evaluate our synthetic glucose traces for use in a glucose forecasting task using the common paradigm TSTR (Train on Synthetic, Test on Real), in which the synthetic data is used to train the model and then tested on the real validation data. We train an LSTM network optimized for glucose forecasting tasks [28] and report the Root Mean Square Error (RMSE) in Table 1. We run the experiment 10 times and train the LSTM for 10,000 epochs. We have also tested with other models including RNNs, attention-based models and other LSTM architectures (such as bidirectional LSTMs) but show the results for the best performing model, the LSTM optimized for glucose forecasting. Since RMSE provides a limited view about the model's predictions, we also plot the Clarke Error Grid [29], which visualizes the differences between a predictive and reference measurement, and is a basis for evaluating the safety of diabetes-related medical devices. More details are in Appendix E. GlucoSynth provides the best forecasting results compared to all other models across all privacy budgets.

Table 1: Fidelity, Breadth and Utility Evaluation. Fidelity: bolded values do not have a statistically significant difference from the real data (what we want). Breadth and Utility: VM = fraction found validation motifs; We want high VM, Coverage and low MSE, RMSE; Bolded values indicate the best ones at each privacy budget (nonprivate compared with private models when $\epsilon = \infty$).

| Model | $\epsilon$ | Fidelity (metric, p-val) Variance | Time-in-Range | Breadth VM | Coverage | MSE | Utility RMSE |
|---|---|---|---|---|---|---|---|
| GlucoSynth | 0.01 | 2576, <$1e-5$ | 61.8, $2e-5$ | **1.000** | 0.010 | **99.0** | **$0.038 \pm 3e-4$** |
| | 0.1 | **2809, 0.356** | **60.1, 0.532** | **1.000** | 0.083 | **11.2** | **$0.036 \pm 3e-4$** |
| | 1 | 2761, 0.022 | **60.6, 0.410** | **0.992** | 0.145 | **6.7** | **$0.030 \pm 1e-4$** |
| | 10 | **2801, 0.316** | **60.2, 0.845** | **1.000** | 0.167 | **5.0** | **$0.029 \pm 1e-4$** |
| | $\infty$ | **2812, 0.503** | **60.2, 0.682** | **0.987** | **0.534** | **1.6** | **$7e-3 \pm 2e-4$** |
| TimeGAN | $\infty$ | 2235, $8e-3$ | **62.3, 0.420** | 0.625 | $6e-3$ | 107.7 | $0.061 \pm 3e-4$ |
| FF | $\infty$ | **2836, 0.902** | 46.6, <$1e-5$ | 0.642 | 0.405 | 2.0 | $0.038 \pm 3e-4$ |
| NVP | $\infty$ | 1789, <$1e-5$ | 65.5, <$1e-5$ | 0.482 | 0.328 | 1.9 | $0.029 \pm 3e-5$ |
| RGAN | 0.01 | 57, <$1e-5$ | 78.8, <$1e-5$ | 0.013 | $1e-3$ | 108.6 | $0.819 \pm 0.010$ |
| | 0.1 | 53, <$1e-5$ | 71.6, $3e-5$ | 0.015 | 0.031 | 107.3 | $0.688 \pm 6e-3$ |
| | 1 | 67, <$1e-5$ | 78.2, <$1e-5$ | 0.015 | 0.033 | 103.3 | $0.651 \pm 0.018$ |
| | 10 | 77, <$1e-5$ | 83.7, <$1e-5$ | 0.017 | 0.053 | 100.3 | $0.619 \pm 0.016$ |
| | $\infty$ | 90, <$1e-5$ | 78.0, <$1e-5$ | 0.026 | 0.091 | 79.6 | $0.460 \pm 0.013$ |
| dpGAN | 0.01 | 451, <$1e-5$ | 95.3, <$1e-5$ | 0.094 | **0.054** | 180.1 | $0.205 \pm 5e-3$ |
| | 0.1 | 1057, <$1e-5$ | 86.4, <$1e-5$ | 0.390 | **0.195** | 28.9 | $0.045 \pm 2e-4$ |
| | 1 | 875, <$1e-5$ | 86.6, <$1e-5$ | 0.480 | **0.239** | 23.2 | $0.030 \pm 2e-5$ |
| | 10 | 1030, <$1e-5$ | 88.1, <$1e-5$ | 0.743 | **0.251** | 16.1 | $0.035 \pm 8e-5$ |
| | $\infty$ | 1121, <$1e-5$ | 81.8, <$1e-5$ | 0.855 | 0.293 | 10.9 | $0.028 \pm 5e-5$ |

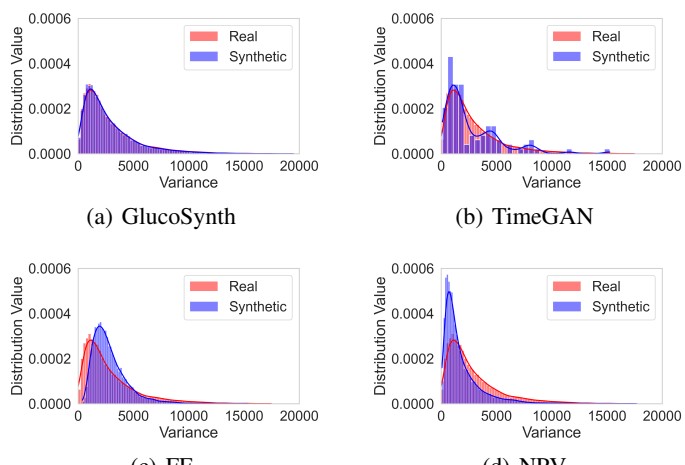

(a) GlucoSynth      (b) TimeGAN

(c) FF      (d) NPV

Figure 6: Distributional Variance for Nonprivate Models

# 8 Limitations & Conclusion

**Limitations.** In order to train on a huge set of glucose traces, we used a private dataset, not publicly available (one of the motivations for this project was actually to share a synthetic version of these traces). That being said, smaller samples of glucose traces with similar patient populations are available at OpenHumans [30] and T1D Exchange Registry [31]. In addition, one of the reasons our privacy results perform well is because we use two *separate* datasets for the training of the motif causality block and the GAN. However, this may be a limiting factor for others that do not have a large enough set of traces available to be able to train adequately on partitioned data.

**Conclusion.** In this paper we have presented GlucoSynth, a novel GAN framework with integrated differential privacy to generate synthetic glucose traces. GlucoSynth conserves motif relationships within the traces, in addition to the typical temporal dynamics contained within time series. We presented a comprehensive evaluation using 1.2 million glucose traces wherein our model outperformed all previous models across three criteria of fidelity, breadth and utility.

## Acknowledgments and Disclosure of Funding

This work was supported in part by the U.S. National Science Foundation under Grant CCF-1942836, CNS-2213700, CNS-2220433, OAC-2319988 and by a Graduate Research Fellowship under Grant No. 1842490. This work was also supported by a Dexcom Graduate Fellowship.

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

# A  Extended Related Work

We overview related work in three lines of research: time series, conditional time series, and time series methods that employ differential privacy. Table 2 summarizes previous time series synthesis methods. We note that there have been exciting developments for adjacent research tasks (data augmentation, forecasting) such as diffusion models [32], but there are not yet any publicly available models specifically for the generation of complete synthetic time series datasets. As such, we focus the scope of our comparison on the current state-of-the-art methods for synthetic time series which all build upon Generative Adversarial Networks (GANs) [3] and transformation-based approaches [9]. In particular TimeGAN [4], RGAN [11] and dpGAN [14] are most similar to ours and used as benchmarks in the evaluation in Section 7.

Table 2: Summary of Previous Methods for Time Series Synthesis. *CI = conditional information or extra features

| Name | Private? | No Labels Required? | No CI*? | Length |
|---|---|---|---|---|
| TimeGAN [4] | x | ✓ | ✓ | 24 - 58 |
| TTS-GAN [5] | x | x | ✓ | 24 - 150 |
| SigCWGAN [33] | x | ✓ | x | 80,000 |
| RGAN [11] | ✓ | ✓ | ✓ | 16 - 30 |
| RCGAN [11] | ✓ | ✓ | x | 16 - 30 |
| dpGAN [14] | ✓ | ✓ | ✓ | 96 |
| RDP-CGAN [6] | ✓ | ✓ | x | 2 - 4097 |
| DoppelGANger  [34] | ✓ | ✓ | x | 50 - 600 |
| GlucoSynth (Ours) | ✓ | ✓ | ✓ | 288 |

**Time Series.** There have been promising models to generate synthetic time series across a variety of domains such as financial data [35], cyber-physical systems (e.g., smart homes [36]), and medical signals [37]. Brophy et al. [10] provides a survey of GANs for time series synthesis. TimeGan [4] is a popular benchmark that jointly learns an embedding space using supervised and adversarial objectives in order to capture the temporal dynamics amongst traces. TTS-GAN [5], trains a GAN model that uses a transformer encoding architecture in order to best preserve temporal dynamics. Transformation-based approaches have also had success for time series data. Real-valued non-volume preserving transformations (NVP) [9] model the underlying distribution of the real data using generative probabilistic modeling and use this model to output a set of synthetic data. Similarly, Fourier Flows (FF) [12] transform input traces into the frequency domain and output a set of synthetic data from the learned spectral representation of the original data. Methods that only focus on learning the temporal or distributional dynamics in time series are not sufficient for generating *realistic* synthetic glucose traces due to the lack of temporal dependence within sequences of glucose motifs.

**Conditional Time Series.** Many works have developed time series models that supplement their training using extra features or conditional data. Esteban et al. [11] develops two GAN models (RGAN/RCGAN) with RNN architectures, conditioned on auxiliary information provided at each timestep during training. SigCWGAN [33] uses a mathematical conditional metric ($Sig - W_1$) characterizing the signature of a path to capture temporal dependence of joint probability distributions in long time series data. However, our glucose traces do not have any additional information available so these methods cannot be used[1].

**Differentially-Private GANs.** To protect sensitive data, several GAN architectures (DP GANs) have been designed to incorporate privacy-preserving noise needed to satisfy differential privacy guarantees [13]. Although DP GANs such as PateGAN [38] have had great success for other data types and learning tasks (e.g., tabular data, supervised classification tasks), results have been less satisfactory in DP GANs developed for time series.

RGAN/RCGAN [11] also includes a DP implementation, but the authors find large gaps in performance between the nonprivate and private models. Frigerio et al. [14] extends a simple DP GAN architecture (denoted dpGAN) to to time-series data. The synthetic data from their private model

---

[1]There is a caveat here that RGAN does not use auxillary information, hence why we compare with it in our benchmarks.

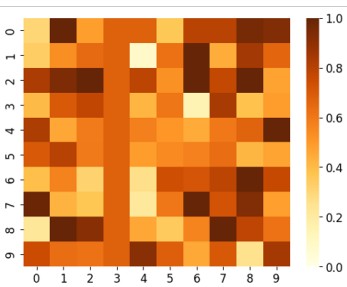

Figure 7: Example motif causality matrix for a small motif set ($m = 10$). Each value in the grid is between 0 and 1. 0 indicates no motif-causal relationship, and 1 indicates the strongest motif causal relationship.

conserves the distribution of the real data but loses some of the variability (diversity) from the original samples. RDP-CGAN [6] develops a convolutional GAN architecture that uses Rényi differential privacy specifically for medical data. Across different datasets, they find that reasonable privacy budgets result in major drops in the performance of the synthetic data. Finally, DoppelGANger [34] develops a temporal GAN framework for time series with metadata and perform an in-depth privacy evaluation. Notably, they find that providing strong theoretical DP guarantees results in destroying the fidelity of the synthetic data, beyond anything feasible for use in real-world scenarios. Each of these methods touches on the innate challenge of generating DP synthetic time series due to very high tradeoffs between utility and privacy. Our DP framework uses two different methods to integrate privacy into our GAN architecture, resulting in a better utility-privacy trade-off than previous methods.

## B    Additional Experimental Details

**Note on Data Use.** As explained in the approach (Section 5), our model uses two *separate* datasets for the training of the motif causality block and the rest of the GAN. As such, we used two different samples of glucose traces with no overlap between patients for the training of each section (meaning we actually used a total of 2.4 million traces across the entire model). We also note that we have received the proper ethical and legal consent from the individuals to use their data in this way (and for this purpose).

**Hyperparameters.** Our experiments were completed in the Google Cloud platform on an Intel Skylake 96-core cpu with 360 GB of memory. We use a separate validation dataset (not the set of original training traces) for all experimental results. Throughout all our experiments we use GlucoSynth model parameters of $\alpha = 0.1$ and $\eta = 10$ and a motif tolerance of $\sigma = 2$ mg/dL and motif length $\tau = 48$. Motif length of 48 timesteps is equivalent to 4 hours of time and represents a clinically significant threshold. This threshold was chosen because the effect of any behaviors on glucose occur within 4 hours of the event (e.g., the effect from eating a meal – a rise in glucose – will occur within 4 hours after eating.) We note that other choices for $\tau$ could be used, based on what types of phenomenon the users wish to replicate; for example, to capture day/night glucose rhythm effects, we suggest a $\tau$ of 144, corresponding to 12 hours of time.

We vary $\epsilon$ in our privacy experiments, but keep $\delta$ the same at $5e{-}4$. Importantly, in order to meet our privacy guarantees, we assume that privacy is provisioned at the trace level and each individual has only one trace in the dataset. The motif set is derived separately from the training data (either from a public dataset or generated based on knowledge about the underlying data, e.g., the possible glucose motif combinations), so as not to effect the differential privacy guarantees or use up any privacy budget. In our case, we assume the motif set is all-encompassing and generated from the universe of possible motifs, resulting in $m = 5,977,610$ total motifs in the motif set.

**Benchmark Details.**    TimeGAN [4] is implemented from `www.github.com/jsyoon0823/TimeGAN`; Fourier Flows (FF) [12] are implemented from `www.github.com/ahmedmalaa/Fourier-flows`; RGAN [11] is implemented from `www.github.com/ratschlab/RGAN`; and DPGAN [14] is adapted from `www.github.com/SAP-samples/security-research-differentially-private-generative-models`. All the benchmarks were trained according to

their suggested parameters, with most models trained for 10,000 epochs. We note that we trained for more than the suggested epochs (50,000 instead of 10,000) and tried many additional hyperparameter settings for RGAN to attempt to improve its performance and provide the fairest comparison possible.

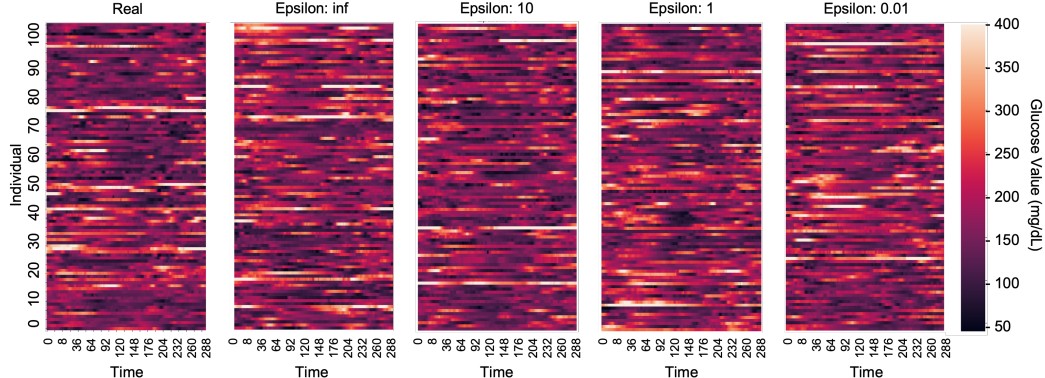

Figure 8: Heatmaps for GlucoSynth Across Different Privacy Budgets

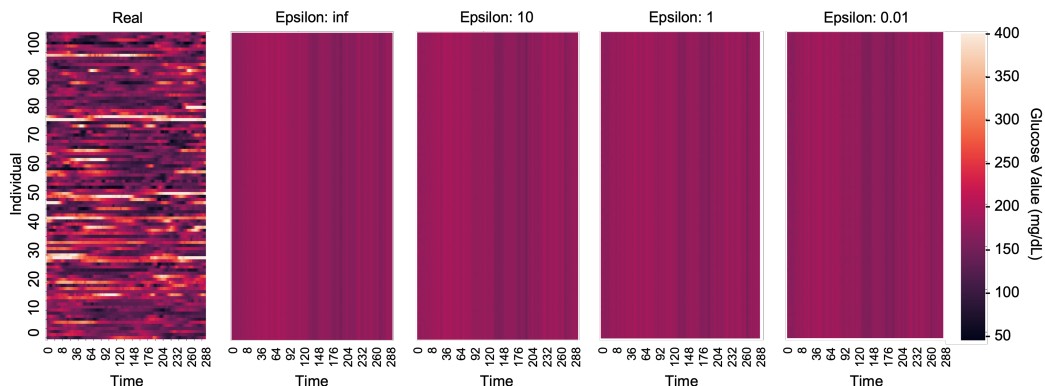

Figure 9: Heatmaps for RGAN Across Different Privacy Budgets

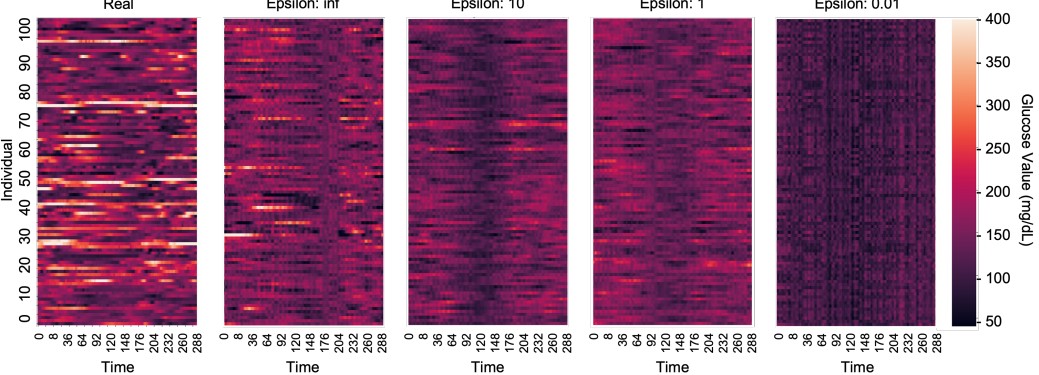

Figure 10: Heatmaps for dpGAN Across Different Privacy Budgets

## C Additional Evaluation: Fidelity

### C.1 Visualizations

We provide heatmap visualizations of sample real and synthetic glucose traces from all the models. Although this is not a comprehensive way to evaluate trace quality, it does give a snapshot view about how the synthetic traces compare to the real ones. Each heatmap contains 100 randomly sampled

glucose traces. Each row is a single trace from timestep 0 to 288. The values (coloring) in each row indicate the glucose value (between 40 mg/dL and 400 mg/dL). Figure 5 shows the nonprivate models, and Figures 8, 9, 10 show the private models with different privacy budgets. Upon examining the heatmaps, we notice that GlucoSynth consistently generates realistic looking glucose traces, even at very small privacy budgets.

## C.2 Population Statistics

In order to evaluate fidelity on a population scale, we compute a common set of glucose metrics used to evaluate patient glycemic control on the real and synthetic data, including average trace variability (VAR), Time in Range (TIR), the percentage of time glucose is within the clinical guided range of 70-180mg/dL; and time hypo- and hyper- glycemic (time below and above range, respectively) in Table 4. More details on each of the metrics are included in Table 3. We test if the difference in metrics between the synthetic and real data is statistically significant, using a p-value of 0.05. A p-value <0.05 indicates the difference is statistically significant. We want synthetic data that has similar population statistics to the real data: p-values > 0.05 such that the differences in statistics between real and synthetic data are not significant. GlucoSynth outperforms all other models, with no statistically significant difference in all metrics for privacy budgets of $\epsilon \geq 100$ and only one metric with a statistically significant difference for budgets $\epsilon = 1 - 10$.

Table 3: Glycemic Metric Explanations

| Metric | Name | Explanation |
|---|---|---|
| VAR | Signal Variance | average trace variability |
| TIR | Time in Range | % of time glucose $\geq 70$ & $\leq 180$ |
| Hypo | Time Hypoglycemic | % of time glucose $< 70$ |
| Hyper | Time Hyperglycemic | % of time glucose $> 180$ |
| GVI | Glycaemic Variability Index | more detailed measure of glucose variability |
| PGS | Patient Glycaemic Status | metric combining GVI and TIR |

Table 4: Population Data Statistics. Each cell value for the synthetic data shows the (metric, p-value) using a 0.05 testing threshold. Bolded values do not have a statistically significant difference from the real data (what we want).

| Model | $\epsilon$ | VAR | TIR | Hypo | Hyper | GVI | PGS |
|---|---|---|---|---|---|---|---|
| Real Data | N/A | 2832.76 | 60.31 | 1.58 | 38.11 | 4.03 | 349.23 |
| GlucoSynth | 0.01 | 2575.501, 0.0 | 61.759, $2.0e-5$ | 1.331, 0.0 | 36.91, $5.66e-4$ | **4.002, 0.085** | 323.056, 0.0 |
| | 0.1 | **2803.513, 0.356** | **60.088, 0.532** | 1.264, 0.0 | **38.648, 0.137** | 3.969, $2.74e-4$ | **347.562, 0.712** |
| | 1 | 2760.853, 0.022 | **60.597, 0.41** | **1.512, 0.163** | **37.892, 0.537** | **4.019, 0.577** | **345.159, 0.368** |
| | 10 | **2800.805, 0.316** | **60.24, 0.845** | **1.538, 0.395** | **38.222, 0.76** | 3.963, $6.7e-5$ | **344.376, 0.28** |
| | 100 | **2796.424, 0.244** | **60.138, 0.625** | **1.567, 0.808** | **38.295, 0.609** | **4.044, 0.32** | **352.679, 0.449** |
| | $\infty$ | **2811.622, 0.503** | **60.165, 0.682** | **1.54, 0.416** | **38.295, 0.61** | **4.056, 0.083** | **353.584, 0.339** |
| TimeGAN | $\infty$ | 2234.576, $8.08e-3$ | **62.315, 0.42** | 0.657, $8.233e-3$ | **37.028, 0.669** | 5.482, 0.0 | 503.148, $0.2e-5$ |
| FF | $\infty$ | **2836.067, 0.902** | 46.578, 0.0 | 5.627, 0.0 | 47.795, 0.0 | 4.931, 0.0 | 528.773, 0.0 |
| NVP | $\infty$ | 1789.430, 0.0 | 65.499, 0.0 | **1.507, 0.154** | 32.994, 0.0 | 6.607, 0.0 | 589.473, 0.0 |
| RGAN | 0.01 | 56.96, 0.0 | 78.756, 0.0 | 0.0, $1.78e-4$ | 21.244, 0.0 | 2.52, 0.0 | 93.409, 0.0 |
| | 0.1 | 52.553, 0.0 | 71.617, $3.7e-5$ | 0.0, $1.78e-4$ | 25.715, 0.0 | 2.208, 0.0 | 98.944, 0.0 |
| | 1 | 67.346, 0.0 | 78.154, 0.0 | 0.0, $1.78e-4$ | 21.846, 0.0 | 2.251, 0.0 | 85.417, 0.0 |
| | 10 | 76.632, 0.0 | 83.681, 0.0 | 0.0, $1.78e-4$ | 16.319, 0.0 | 2.23, 0.0 | 64.562, 0.0 |
| | 100 | 84.918, 0.0 | 74.285, 0.0 | 0.0, $1.78e-4$ | 25.715, $0.6e-5$ | 2.208, 0.0 | 98.944, 0.0 |
| | $\infty$ | 89.702, 0.0 | 78.044, 0.0 | 0.0, $1.78e-4$ | 21.956, 0.0 | 2.184, 0.0 | 82.923, 0.0 |
| dpGAN | 0.01 | 451.098, 0.0 | 95.275, 0.0 | 4.60, 0.0 | 0.124, 0.0 | 7.718, 0.0 | 41.549, 0.0 |
| | 0.1 | 1057.205, 0.0 | 86.43, 0.0 | 0.837, 0.0 | 12.732, 0.0 | 6.349, 0.0 | 148.412, 0.0 |
| | 1 | 874.663, 0.0 | 86.631, 0.0 | 1.135, 0.0 | 12.234, 0.0 | 4.794, 0.0 | 118.286, 0.0 |
| | 10 | 1029.971, 0.0 | 88.122, 0.0 | 2.002, 0.0 | 9.876, 0.0 | 4.759, 0.0 | 93.632, 0.0 |
| | 100 | 821.636, 0.0 | 89.354, 0.0 | 0.664, 0.0 | 9.982, 0.0 | 4.613, 0.0 | 82.561, 0.0 |
| | $\infty$ | 1120.553, 0.0 | 81.773, 0.0 | 1.359, $0.3e-5$ | 16.868, 0.0 | 6.248, 0.0 | 188.991, 0.0 |

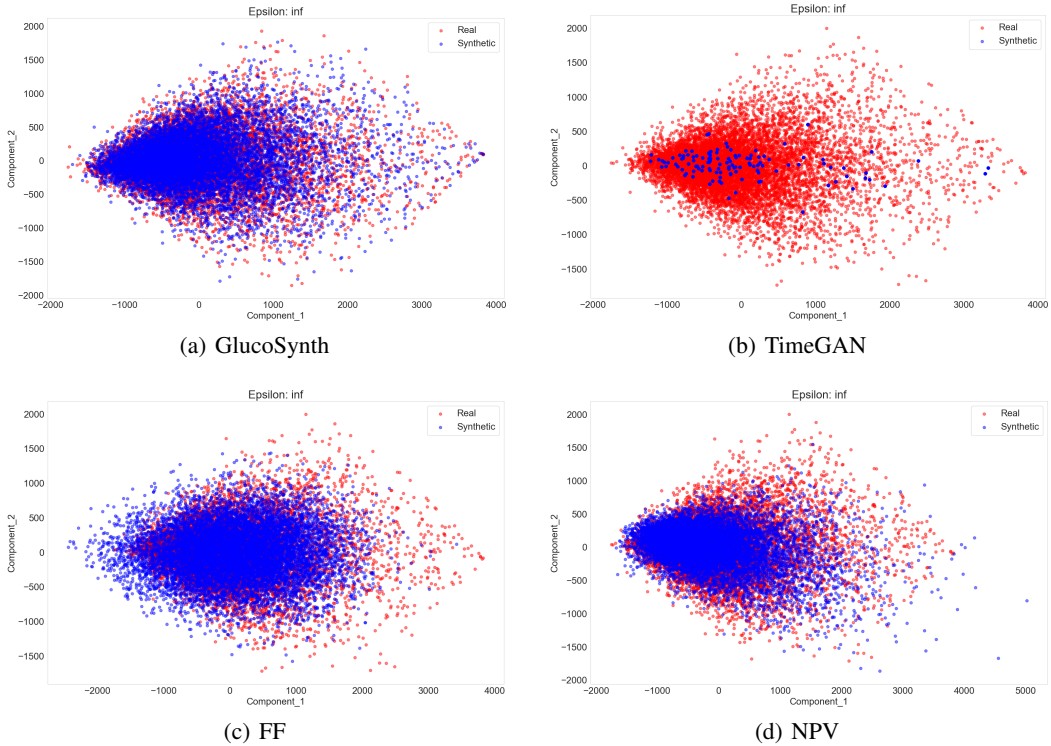

Figure 11: PCA Comparison for Nonprivate Models

## C.3 Distributional Comparisons

We visualize differences in distributions between the real and synthetic data by plotting the distribution of variances and using PCA [27]. Figure 6 and Figure 11 show the variance distribution and PCA plots, respectively for the nonprivate models. We also compare distributional changes across privacy budgets: Figures 12 and 13 show GlucoSynth, Figures 14 and 15 show RGAN and Figures 16 and 17 show dpGAN.

Looking at the figures, GlucoSynth better captures the distribution of the real data compared to all of the nonprivate models. As evidenced in the PCA plot, (Fig. 11), FF comes the closest to capturing the real distribution in its synthetic data, but ours does a better job of representing the more rare types of traces. GlucoSynth also outperforms all of the private models across all privacy budgets. Even at small budgets ($\epsilon < 1$), the general shape of the overall distribution is conserved (e.g., see Figure 12).

# D   Additional Evaluation: Breadth

Compared to all other models across all privacy budgets, our model has the best ratio of found validation motifs, with close to 1.0 for VM and the lowest MSEs. It also has the best coverage for nonprivate settings and an $\epsilon$ of 100. Interestingly, dpGAN has the best coverage compared to all other models for privacy budgets $\epsilon \leq 10$ but worse MSEs across all budgets than GlucoSynth. This means that although it finds a broader *number* of motifs contained in the real data, the overall distributions of motifs it creates in the synthetic data have much higher error rates. We argue that the tradeoff found by our model is better because although it does miss some of the *types* of motifs from the real data (misses some breadth), from the ones it does find it constructs realistic distributions of the motifs and generates very few anomalous ones.

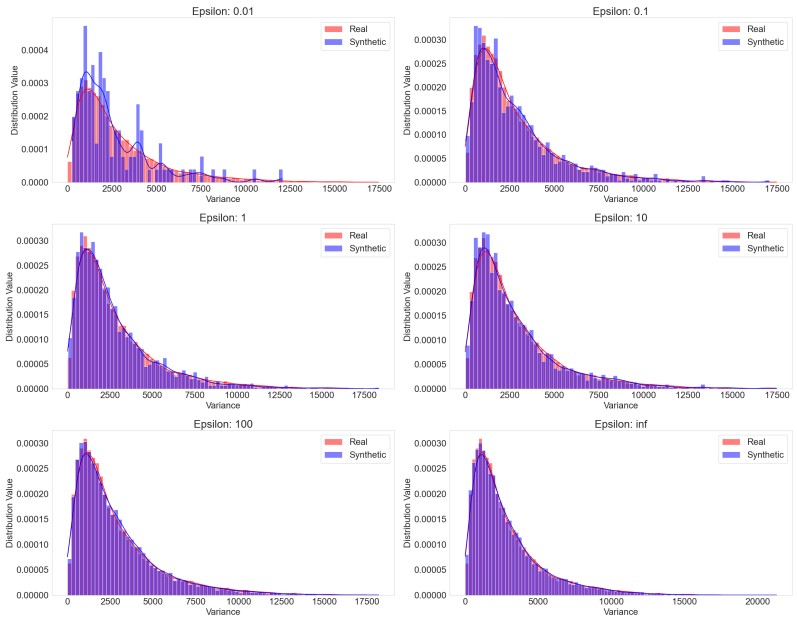

Figure 12: GlucoSynth Distributional Variance Comparison Across Privacy Budgets

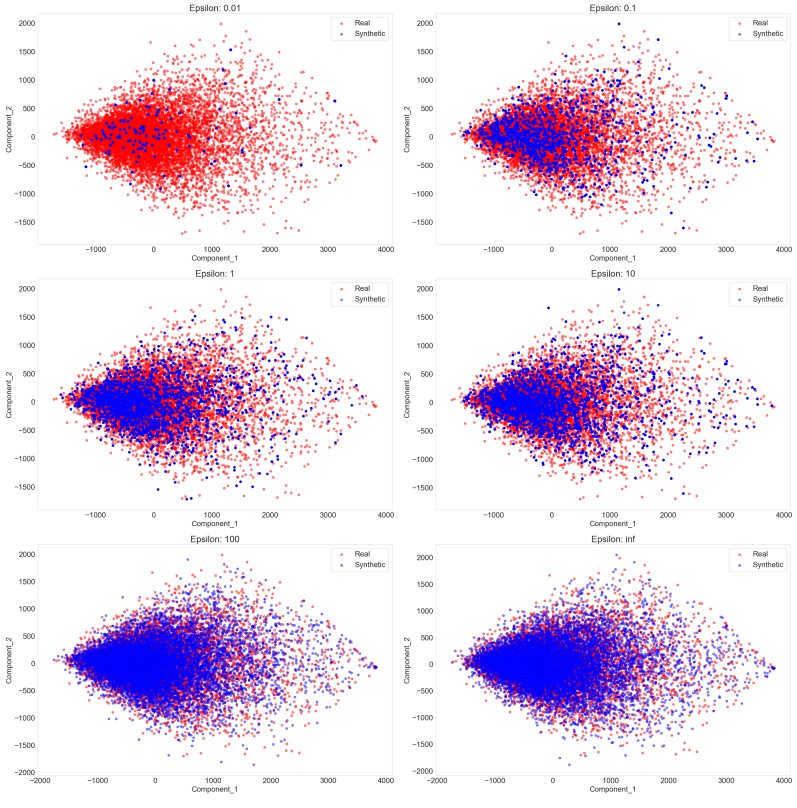

Figure 13: GlucoSynth PCA Comparison Across Privacy Budgets

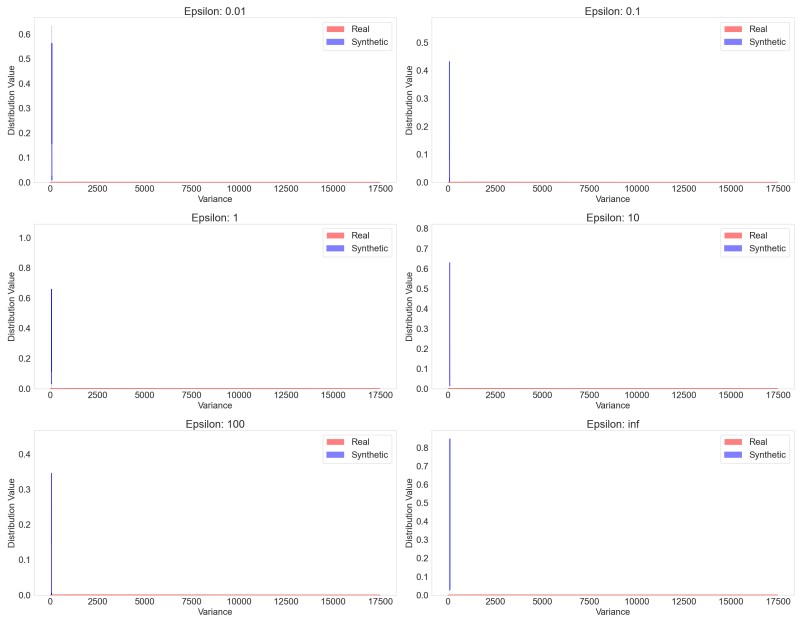

Figure 14: RGAN distributional Variance Comparison Across Privacy Budgets

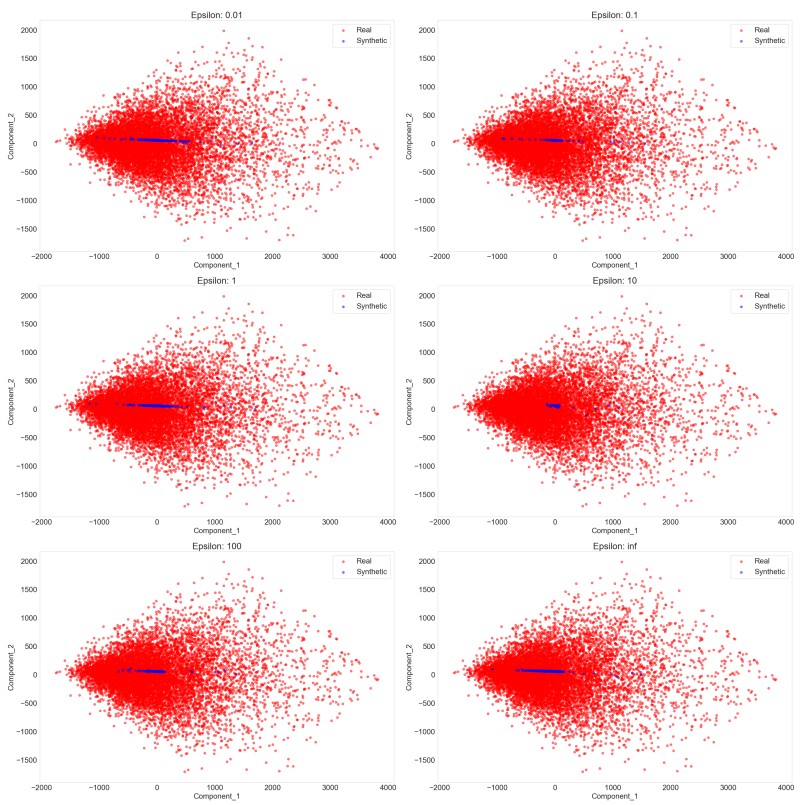

Figure 15: RGAN PCA Comparison Across Privacy Budgets

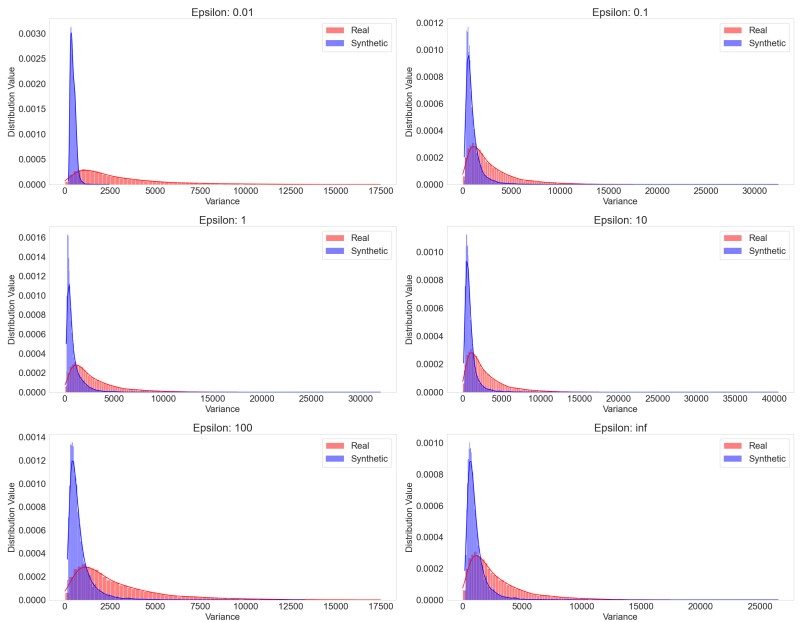

Figure 16: dpGAN distributional Variance Comparison Across Privacy Budgets

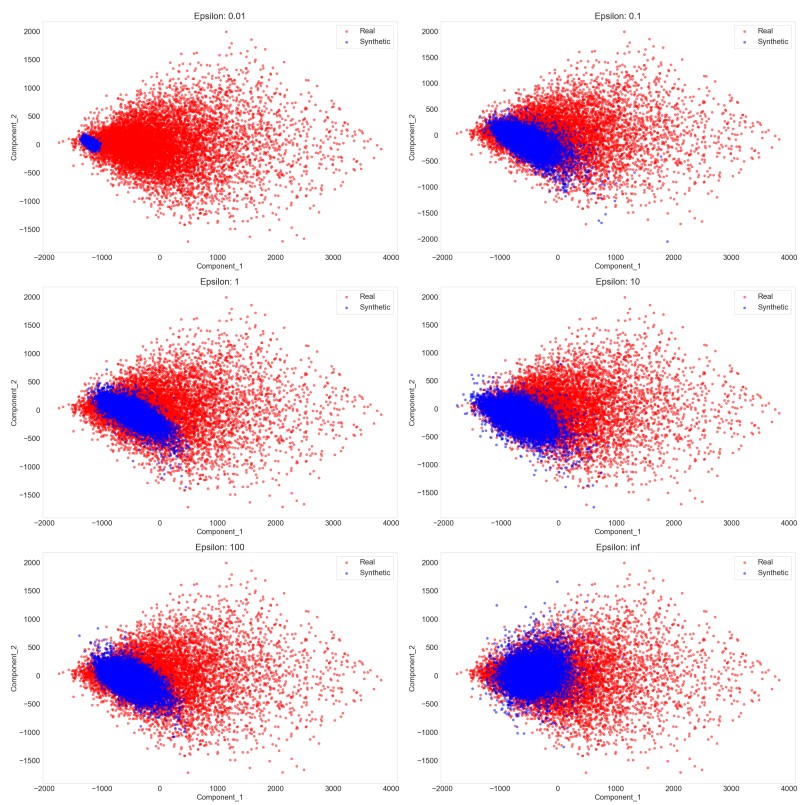

Figure 17: dpGAN PCA Comparison Across Privacy Budgets

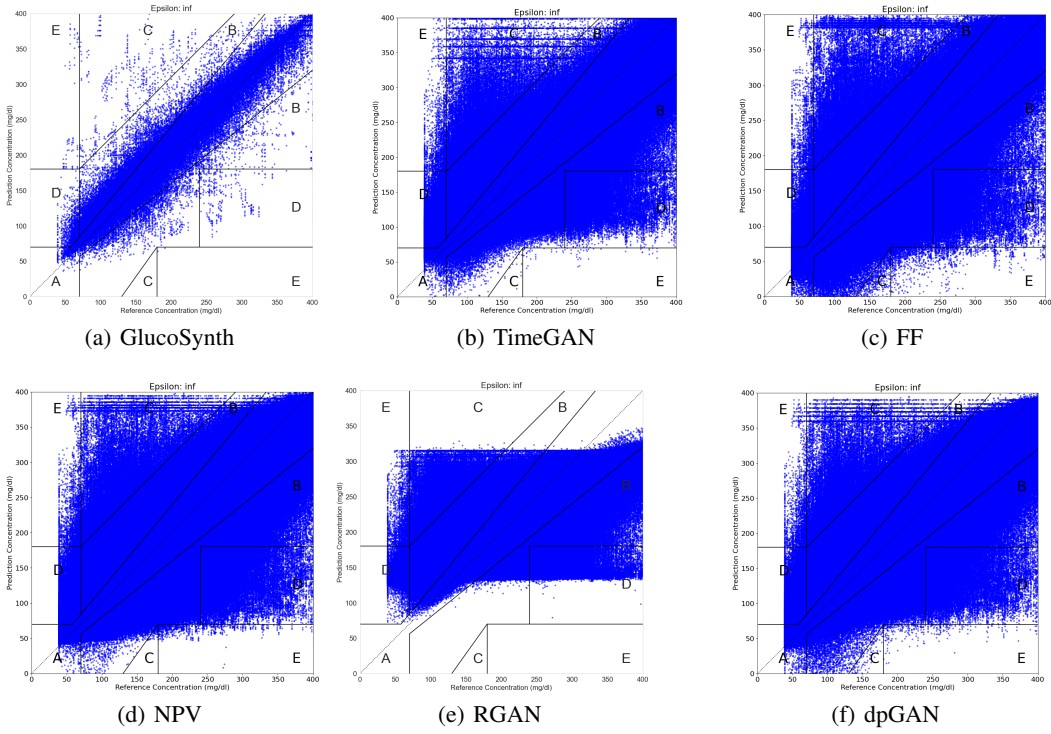

| (a) GlucoSynth | (b) TimeGAN | (c) FF |
|:---:|:---:|:---:|

| (d) NPV | (e) RGAN | (f) dpGAN |
|:---:|:---:|:---:|

Figure 18: Clarke Error Zone Figures for All Models

# E   Additional Evaluation: Utility

Since RMSE may provide a limited view about the predictions from the glucose forecasting model, we also plot the Clarke Error Grid [29], which visualizes the differences between a predictive measurement and a reference measurement, and is the basis used for evaluation of the safety of diabetes-related medical devices (for example, used for evaluating glucose outputs from predictive models integrated into artificial insulin delivery systems). The Clarke Error Grid is implemented using `www.github.com/suetAndTie/ClarkeErrorGrid`. The grids are shown in Figure 18.

In the figures, the x-axis is the reference value and the y-axis is the prediction. A diagonal line means the predicted value is exactly the same as the reference value (the best case). There are 5 total zones that make up the grid, listed in order from best to worst:

- Zone A – Clinically Accurate: Predictions differ from actual values by no more than 20% and lead to clinically correct treatment decisions.
- Zone B – Clinically Acceptable: Predictions differ from actual values by more than 20% but would not lead to any treatment decisions.
- Zone C – Overcorrections: Acceptable glucose levels would be corrected (overcorrection).
- Zone D – Failure to Detect: Predictions lie within the acceptable range but the actual values are outside the acceptable range, resulting in a failure to detect and treat errors in glucose.
- Zone E – Erroneous Treatment: Predictions are opposite the actual values, resulting in erroneous treatment, opposite of what is clinically recommended.

We show Clarke Error grids for all models (and the private models with no privacy included, $\epsilon = \infty$). This is because comparing the models at different privacy budgets is not very informative – it can be hard to tell exactly where changes between different budgets may occur. We also present a table with the percentages of predicted datapoints in each category in Table 5. This table includes a comparison among different privacy budgets for the private models (much more effective than the figures by themselves.)

Table 5: Clarke Error Grid Zones. Value is the percentage of predicted datapoints. Categories go from A to E, best to worst. Bolded rows indicate the best results on the synthetic data at each privacy budget (nonprivate models compared with private models when $\epsilon = \infty$)

| Model | $\epsilon$ | A: Accurate | B: Acceptable | C: Overcorrection | D: Failure to Detect | E: Error |
|-------|-----|-------------|---------------|-------------------|----------------------|----------|
| GlucoSynth | 0.01 | **0.858 ± 1.057e−3** | **0.131 ± 1.172e−3** | **3.271e−3 ± 0.0** | **0.017 ± 1.158e−4** | **5.79e−6 ± 1.2e−6** |
| | 0.1 | **0.863 ± 6.947e−3** | **0.126 ± 7.526e−4** | **3.054e−3 ± 1.45e−5** | **0.018 ± 4.34e−5** | **5.79e−6 ± 0.0** |
| | 1 | **0.862 ± 1.578e−3** | **0.128 ± 1.259e−3** | **3.343e−3 ± 1.45e−5** | **0.016 ± 3.329e−4** | **5.79e−6 ± 0.0** |
| | 10 | **0.864 ± 6.947e−3** | **0.125 ± 6.513e−4** | **3.039e−3 ± 5.79e−5** | **0.017 ± 4.34e−5** | **8.68e−6 ± 2.89e−5** |
| | 100 | **0.864 ± 1.74e−3** | **0.126 ± 1.447e−3** | **3.387e−3 ± 0.0** | **0.017 ± 2.895e−4** | **5.79e−6 ± 0.0** |
| | $\infty$ | **0.964 ± 1.201e−3** | **0.035 ± 1.158e−3** | **3.039e−4 ± 2.89e−5** | **1.732e−4 ± 1.158e−4** | **8.68e−6 ± 1.45e−5** |
| TimeGAN | $\infty$ | 0.741 ± 0.012 | 0.233 ± 0.012 | 2.240e−3 ± 9.8e−5 | 0.024 ± 8.44e−4 | 2.19e−4 ± 1.9e−5 |
| FF | $\infty$ | 0.824 ± 6.624e−3 | 0.156 ± 6.148e−3 | 3.547e−3 ± 9.0e−5 | 0.017 ± 3.940e−4 | 3.57e−4 ± 8.0e−6 |
| NVP | $\infty$ | 0.79 ± 3.03e−4 | 0.186 ± 3.87e−4 | 3.49e−3 ± 1.5e−5 | 0.02 ± 1.04e−4 | 3.58e−4 ± 5.0e−6 |
| RGAN | 0.01 | 0.54 ± 0.014 | 0.435 ± 0.014 | 3.389e−4 ± 1.197e−4 | 0.024 ± 2.71e−4 | 2.429e−4 ± 3.43e−5 |
| | 0.1 | 0.594 ± 1.998e−3 | 0.38 ± 1.74e−3 | 1.326e−3 ± 1.429e−4 | 0.025 ± 1.069e−4 | 2.873e−4 ± 8.68e−6 |
| | 1 | 0.637 ± 6.785e−3 | 0.336 ± 6.128e−3 | 2.661e−3 ± 1.87e−5 | 0.024 ± 6.464e−4 | 2.792e−4 ± 2.95e−5 |
| | 10 | 0.634 ± 3.452e−3 | 0.338 ± 3.247e−3 | 2.253e−3 ± 1.004e−4 | 0.025 ± 2.894e−4 | 3.027e−4 ± 1.71e−5 |
| | 100 | 0.638 ± 4.709e−3 | 0.335 ± 4.219e−3 | 1.991e−3 ± 2.17e−5 | 0.025 ± 4.884e−4 | 2.949e−4 ± 2.26e−5 |
| | $\infty$ | 0.646 ± 6.89e−4 | 0.326 ± 7.19e−4 | 2.613e−3 ± 2.852e−4 | 0.024 ± 3.006e−4 | 2.859e−4 ± 1.5e−5 |
| dpGAN | 0.01 | 0.308 ± 3.482e−3 | 0.509 ± 3.71e−3 | 2.894e−7 ± 0.0 | 0.183 ± 2.33e−4 | 1.114e−5 ± 4.196e−6 |
| | 0.1 | 0.781 ± 6.35e−4 | 0.191 ± 5.37e−4 | 3.226e−3 ± 5.715e−5 | 0.024 ± 3.8e−5 | 2.533e−4 ± 1.881e−6 |
| | 1 | 0.786 ± 5.44e−4 | 0.187 ± 5.81e−4 | 2.409e−3 ± 2.894e−7 | 0.024 ± 3.6e−5 | 2.078e−4 ± 5.787e−7 |
| | 10 | 0.806 ± 7.34e−4 | 0.169 ± 6.09e−4 | 2.386e−3 ± 1.476e−5 | 0.023 ± 1.113e−4 | 2.146e−4 ± 2.749e−6 |
| | 100 | 0.813 ± 3.18e−4 | 0.161 ± 2.86e−4 | 2.266e−3 ± 2.083e−5 | 0.023 ± 5.4e−5 | 1.889e−4 ± 1.013e−6 |
| | $\infty$ | 0.819 ± 1.487e−3 | 0.16 ± 1.306e−3 | 3.193e−3 ± 2.677e−5 | 0.018 ± 1.60e−4 | 3.166e−4 ± 5.208e−6 |

Looking at the grids, we can see that GlucoSynth performs the best, with most of the values along the diagonal axis (Zone A and B) and less around the other zones (Zones C-E) as compared to the other models. This means that most of the predicted glucose values from the model trained on our synthetic data are in the Clinically Accurate and Acceptable ranges, with less in the erroneous zones. Moreover, by examining the table we see that GlucoSynth outperforms all other models across all privacy budgets as well.

