# GlucoSynth: Generating Differentially-Private Synthetic Glucose Traces

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

|---|---|---|---|---|---|---|---|
| | | Variance | Time-in-Range | VM | Coverage | MSE | RMSE |
| GlucoSynth | 0.01 | 2576, <1e−5 | 61.8, 2e−5 | **1.000** | 0.010 | **99.0** | **0.038 ± 3e−4** |
| | 0.1 | **2809, 0.356** | 60.1, 0.532 | **1.000** | 0.083 | **11.2** | **0.036 ± 3e−4** |
| | 1 | 2761, 0.022 | **60.6, 0.410** | 0.992 | 0.145 | **6.7** | **0.030 ± 1e−4** |
| | 10 | **2801, 0.316** | **60.2, 0.845** | **1.000** | 0.167 | **5.0** | **0.029 ± 1e−4** |
| | ∞ | **2812, 0.503** | **60.2, 0.682** | 0.987 | **0.534** | **1.6** | **7e−3 ± 2e−4** |
| TimeGAN | ∞ | 2235, 8e−3 | **62.3, 0.420** | 0.625 | 6e−3 | 107.7 | 0.061 ± 3e−4 |
| FF | ∞ | **2836, 0.902** | 46.6, <1e−5 | 0.642 | 0.405 | 2.0 | 0.038 ± 3e−4 |
| NVP | ∞ | 1789, <1e−5 | 65.5, <1e−5 | 0.482 | 0.328 | 1.9 | 0.029 ± 3e−5 |
| RGAN | 0.01 | 57, <1e−5 | 78.8, <1e−5 | 0.013 | 1e−3 | 108.6 | 0.819 ± 0.010 |
| | 0.1 | 53, <1e−5 | 71.6, 3e−5 | 0.015 | 0.031 | 107.3 | 0.688 ± 6e−3 |
| | 1 | 67, <1e−5 | 78.2, <1e−5 | 0.015 | 0.033 | 103.3 | 0.651 ± 0.018 |
| | 10 | 77, <1e−5 | 83.7, <1e−5 | 0.017 | 0.053 | 100.3 | 0.619 ± 0.016 |
| | ∞ | 90, <1e−5 | 78.0, <1e−5 | 0.026 | 0.091 | 79.6 | 0.460 ± 0.013 |
| dpGAN | 0.01 | 451, <1e−5 | 95.3, <1e−5 | 0.094 | **0.054** | 180.1 | 0.205 ± 5e−3 |
| | 0.1 | 1057, <1e−5 | 86.4, <1e−5 | 0.390 | **0.195** | 28.9 | 0.045 ± 2e−4 |
| | 1 | 875, <1e−5 | 86.6, <1e−5 | 0.480 | **0.239** | 23.2 | 0.030 ± 2e−5 |
| | 10 | 1030, <1e−5 | 88.1, <1e−5 | 0.743 | **0.251** | 16.1 | 0.035 ± 8e−5 |

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

 a motif tolerance $\sigma = 2$ mg/dL, motif length $\tau = 48$, and GlucoSynth model parameters of $\alpha = 0.1$ and $\eta = 10$. Motif length of 48 timesteps is equivalent to 4 hours of time; this threshold was chosen because the effect of any behaviors on glucose occur within 4 hours of the event (e.g., the effect from eating a meal – a rise in glucose – will occur within 4 hours after eating.) There are $m = 5,977,610$ total motifs in the motif set. We vary $\epsilon$ in our privacy experiments, but keep $\delta$ the same at $5e-4$. All the benchmarks were trained according to their suggested parameters, with most models trained for 10,000 epochs. We note that we trained for more than the suggested epochs (50,000 instead of 10,000) and tried many additional hyperparameter settings for RGAN to attempt to improve its performance and provide the fairest comparison possible. Our experiments were completed in the Google Cloud platform on an Intel Skylake 96-core cpu with 360 GB of memory.

## C  Additional Evaluation: Fidelity

### C.1  Visualizations

**Traces.** We provide visualizations of sample real and synthetic glucose traces from all the models. Although this is not a comprehensive way to evaluate trace quality, it does give a snapshot view about what synthetic traces may look like. Figure 7 shows randomly sampled individual traces across the

 nonprivate models, and Figure 8 shows traces across different privacy budgets for the private models.
As evidenced by the figures, GlucoSynth produces highly realistic synthetic glucose traces, even at
small privacy budgets.

**Heatmaps.** We also provide a heatmap visualization of the traces, to give a slightly larger snapshot