# OpenReview forum: "GlucoSynth: Generating Differentially-Private Synthetic Glucose Traces"
_NeurIPS.cc/2023/Conference — NeurIPS 2023 poster_

### Official Review · Reviewer_Eaai · 2023-06-23

**Soundness:** 4 excellent
**Presentation:** 4 excellent
**Contribution:** 3 good
**Rating:** 6
**Confidence:** 3

**Summary:**

Instead of the existing DP GAN framework, which cannot capture the relationship in time-series data well when temporal length is not the only relationship but also different event types and do not perform well when more strict DP (differential privacy) condition is required, the paper proposes new architecture based on the glucose trace use-case which overcomes existing weakness by learning and generating (from the GAN generator) at the space of (Granger) causality matrix instead of the data level. From this, the GAN architecture must be adapted accordingly: autoencoder and discriminator are straightforward (autoencoder to encode time-series datapoint into a vector and discriminator looks at two points in this space), but the generator now can generate more realistic data with not just random noise but also the causality matrix which is learned separately.

**Strengths:**

- The application is practical (healthcare), but may possibly be useful for other similar applications with time-series data.
- The modeling of time-series relationship is more general than previous works (not just time but other factors such as different type of events), which is appropriate for application such as healthcare
- Seems already optimized with previous DP work -- with PATE and autoencoder for GAN.
- Extensive evaluation of synthetic data quantitatively and then qualitatively in Appendix.

**Weaknesses:**

- The result and motivation of the work is from a specific motivation. While this means it's practical, we wouldn't know how well it generalizes (though I think it may).
- I am not sure if differential privacy is really at the user-level (as is worded in line 268) or trace/motif/datapoint level (e.g. when one user gives many traces contributing to the dataset). If not a user-level, then the result is not DP with respect to user privacy. I wonder what would happen if dataset is weighted to account for this?
- It is noted from the author that we cannot reproduce the result for private data. This is something the author acknowledged due to privacy of data; something we have to accept. You said "Reproducibility: Yes; although we are unable to share the code and data, we have linked to similar public datasets and provide detailed descriptions of the architecture, its implementation and training regime. Therefore reproducing the results is straightforward.", but shouldn't there then be an evaluation part using these public data then?

Small comments:
- I wish more explanation in comparing the architecture (not pro and con) of this work compared to existing work is added for clearer visibility of what is novel vs adapted from previous work at the model design. It seems related work section only has the pro/con part. In Appendix A it has more information but not specific: e.g. line 507-509 what are these "two methods"? What's the main difference compared to each of [4] [11] and [14] in line 469? What's new compared to [6] (the architecture looks similar)?
- Please make font size of plots in page 16 and after larger. Some are unreadable.

**Questions:**

- Usually I think of generator's training in GAN as getting feedback from discriminator. In line 218-240, this is different: is generator trained directly with these information to warm start? Is this novel or similar to some previous work?
- Similar to Section 5.2 -- is this framework to balance two losses new or similar to some existing work?
For the above: it should be more clear how your architecture is different from existing work, e.g. yours looks similar to [6]. In the related work it only lists how yours compare performance wise, but not architecture or novelty-wise.
- Why do we only pick 4 hours (line 187)? It makes sense for meal-effect, but this would lose the day-rhythm effect such as sleep time, right?
- Why is the fidelity not in increasing order as $\epsilon$ increases in Figure 5, especially for dpGAN? This seems odd.
- Why not compare to [6] as well? Table 2 suggests [6] uses extra features, though the paper [6] https://people.cs.vt.edu/~reddy/papers/IS22.pdf said it can work for unsupervised data as well -- could you explain more?
- Any upperbound in Table 2 of length on your method (and go infinite and others can't?)?
- How to reproduce others' methods? I think the hyperparameter and/or architecture details should be added to Appendix B.


**Limitations:**

- Authors noted about reproducibility issue of private data (line 350-356)
- Have the owner (patients) of these traces given permission to use their data in this way? Given this is a sensitive data (healthcare), I think it's a good practice to add how this data is ethically appropriate to use (e.g. how we have the permission legally, etc.) I am not an expert and this is not to prevent this kind of work (as this is why we have private synthetic data -- for utilizing sensitive data), but I just want to make sure about this.

---

> ### Author Rebuttal · Authors · 2023-08-04
>
> We thank the reviewer for their time and review. Responding to Weakness 2 about DP: This is a good point. The privacy is at the trace level, which we claimed was user-level because we assume that each patient has only one trace in the dataset. We will add a clarifying statement about this assumption in Section 6.
>
> Responding to Weakness 3 about reproducability: This is a good point and we thank the reviewer for this suggestion.
>
> Responding to Small Comments: Thanks for this constructive feedback. We will add more details about the network architectures in the Related Work and increase the plot sizes in the Appendix.
>
> Responding to Question 1: Yes, in addition to the extra information it receives from the other parts (stepwise, motif causality and distributional) the generator also receives the traditional discriminator feedback (see line 253). The point of these other losses is to allow the generator to learn to conserve temporal, motif causal and distributional characteristics of the data. Without them the generator does not converge and/or does not produce realistic glucose sequences. The stepwise and distributional losses are similar in ideology to other previous works, i.e., see TimeGAN [4] and RGAN [11] (though they do not use the same architecture / network configuration that we do), and the motif causality part is novel to our work.
>
> Responding to Question 2: Thanks for this constructive question. Although some other methods balance losses that are similar in ideology to ours, none of them use motif causality. This is an important contribution because without the motif causality part the other models are not able to generate realistic synthetic glucose sequences since they only focus on conserving temporal and distributional characteristics of the data, which do not adequately represent event-driven sequences.
>
> Responding to Question 3: We provide additional justification for the hyperparameter selection (including for $\tau$) in the Appendix B. $\tau = 48$ was chosen because the effect of most behaviors on glucose occur within 4 hours of the event (e.g., as the reviewer mentioned eating as well as things like stress or exercise). The reviewer is correct that day rhythm effects would not be captured in this shorter timeframe, and other sizes of $\tau$ could be selected based on what types of phenomenon the users wish to replicate.
>
> Responding to Question 4: Is the reviewer talking about the Time-in-Range metric in Table 1? If so, Time-in-Range is the percentage of time a patient's glucose is in the proper range (between 70 and 180 mg/dL), and gives a measure of how well controlled an individual is. So a value of 61.8 means that 61.8\% of the time, the patient had their glucose in the proper range. For individuals with T1D, it is really difficult to keep their glucose in the proper range 100\% of the time, and actually most "well-controlled" individuals have time in ranges of 60-70\%. As such, the synthetic data that has Time in ranges of 80-90\% as in dpGAN for the lower privacy budgets are highly unrealistic. As the budgets increase and the time in range decreases, this is getting more and more realistic (closer to the 60-70\% mark.)
>
> Responding to Question 5: Yes, this is true that RDP-CGAN [6] does work without labels (unsupervised) but it is for multivariate data and relies on learning the correlations amongst multiple features. Since our model is for univariate data with only one feature (glucose), we did not feel this would provide a fair comparison (as there would be no other features for RDP-CGAN to learn from).
>
> Responding to Question 6: Thanks for this important question. We are currently experimenting to find what the maximum length that GlucoSynth can handle may be. In the meantime, we will remove the "+" from Table 2 since we do not know the upperbound yet.
>
> Responding to Question 7: Thanks for this constructive comment. We trained all other methods for 10,000 epochs and using the suggested hyperparameters provided by their individual publications. We will add more explicit details about this in Appendix B.
>
> Responding to the last limitation: Thanks for bringing up this important point. Yes, we have received the proper ethical and legal consent from the individuals to use their data in this way (and for this purpose).

---

> > ### Comment · Reviewer_Eaai · 2023-08-20
> >
> > Overall comments from all discussions:
> > - On DP (from reviewer Q4qV): I think DP explanation part is ok, though of course more details could help.
> > - On synthetic data evaluation (from reviewer VFZB): synthetic data evaluation is hard and I agree that human may not help much. Many times the methods seem ad-hoc but this paper already did a bit more than usual I think.
> > - On model architecture scaling (from reviewer 2Sh2): I saw this autoencoder + GAN similar architecture for DP synthetic data generation before (https://arxiv.org/abs/1912.03250 Differentially Private Synthetic Mixed-Type Data Generation For Unsupervised Learning), and it seems to work well for moderate-dimensional data. I think I am ok that it may not scale to high-dim regime as this DP synthetic data is probably hard, and this paper is probably more focused on other contribution such as new way to apply DP in generator space of casuality.
> >
> > Specific reply to authors' comment on my view:
> > - Weakness 3 on reproducibility: what's your plan on this? Do you think it's good to do on some public data as toy example? I think that would make the experiment part stronger.
> > - Question 2: can you add a bit more explanation about yours vs [6] then, e.g. that you use similar loss or some similar ideas, etc, and why [6] itself is not sufficient?
> >
> > Thanks for other answers. They are helpful. Please revise the paper based on what you said in Weakness 2, Questions 6 and 7.
> >
> > Based on every discussion available, I maintain my score and still vote for acceptance of this paper.

---

> > > ### Author Response · Authors · 2023-08-20
> > >
> > > Thanks for your comments and follow up. We will make those revisions.
> > >
> > > Follow up to Weakness 3: Yes, we agree. We plan to train the model on a small dataset of real patients from the T1D exchange and evaluate its performance across the same criteria of fidelity, breadth and utility.
> > >
> > > Follow up to Question 2: In terms of similarities, we both use autoencoders that minimize some sort of reconstruction loss, however the architectures and purposes of each model are very different. In [6], the point of the autoencoder is to encode and preserve correlations amongst multiple features which the generator uses to synthesize high quality multivariate data that maintains relationships amongst different features. The autoencoder uses a convolutional network structure and the loss function is the Binary Cross Entropy between the real data and reconstructed data (data fed through encoder and decoder).
> > >
> > > Alternatively, since we are only worried about a single feature (univariate data) our auto encoder uses an LSTM or RNN architecture and uses a loss function of mean square error of the real data and the reconstructed data. Using the autoencoding architecture from [6] would not be helpful for our generator since we do not care about finding or preserving relationships amongst multiple features, and instead wish to identify which parts (timesteps) of the univariate time series are most important for the generator to preserve.

---

### Official Review · Reviewer_2Sh2 · 2023-06-27

**Soundness:** 2 fair
**Presentation:** 3 good
**Contribution:** 2 fair
**Rating:** 5
**Confidence:** 3

**Summary:**

This paper proposes a DP generation model, i.e. GlucoSynth, particularly for glucose data. To address a few challenges that prior works fail in, GlucoSynth proposes a causality learning block, to learn the dependency between motifs, rather than merely learning the temporal dependencies. The generator is basically a GAN combined with autoencoder. By training GAN and autoencoder with DP-SGD, training causality block with PATE, GlucoSynth is claimed to achieve DP guarantees.

**Strengths:**

1. The problem is well-motivated, and the summary of challenges in prior works is clear.
2. The idea of motif causality block is somehow novel, compared to prior works. (But I did not read too much related works in synthesizing glucose data, so this point is based on the summary in this work.)

**Weaknesses:**

1. Although Sec. 1 and 2 are clear, starting from Sec. 3, the writing is sometimes confusing. See my questions below for details.
2. This method is not neat. Figure 3 clearly shows that you need to train a total of $m\times r$ neural networks for the motif causality block, let alone the autoencoder and GAN in the next step. Given that the GAN and autoencoder are trained with DP-SGD, which introduces notorious time overhead due to the gradient perturbation, this method is both time- and space-inefficient.
3. The main components in GlucoSynth lack proper explanation. For example,
   + Why do you need to build motif networks for each motif? Can you build only one for all motifs?
   + Why do you need an autoencoder? Do you have the ablation study on the autoencoder?
4. In Line 223 and Figure 4, it seems that the generator needs real data as input to "generate" something. How can you use the generator in reality?
5. The DP mechanisms applied in this work are not novel. Both DP-SGD and PATE are existing standard techniques to retain DP.

**Questions:**

1. What do different colors mean in Fig. 1 and 2?
2. I don't get what Fig. 2 intends to mean. What is the relationship between motif 1 and 2? What should we expect to see from the radial graph?
3. It is better to give full descriptions in the caption of figures, e.g. figure 3 and 4.
4. Line 229, how is $M_{\hat{x}}$ calculated from $M$ and $\hat{x}$?
5. What is the difference between MSE loss in Line 217 and 225?

**Limitations:**

This paper has discussed limitations in the end.

---

> ### Author Rebuttal · Authors · 2023-08-04
>
> We thank the reviewer for their time and review. Responding to Weakness 2: The reviewer is correct that our model is time and memory-intensive. Even so, since you only have to train the model once and then can use the produced synthetic data over and over, we believe this is an okay trade-off. We are also currently exploring methods to optimize the model to make it more efficient.
>
> Responding to Weakness 3(a) about motif networks: Thanks for the opportunity to clarify this important point. We build a motif network for each individual motif in order to learn an explicit mapping between all the other motifs and their predictiveness to a particular motif. If we trained all the motifs together using a single motif network, we would not be able to quantify the exact causal effects between each individual motif because we would not know which exact motifs helped with the prediction (only that there is some combination of unknown motifs that contribute to an accurate prediction of a particular motif.) By training each motif network separately, we are able to learn and quantify the exact effect each motif has on each other, without any confounding effects from other motifs.
>
> Responding to Weakness 3(b) about the autoencoder: We did do an ablation study for the autoencoder, and without the autoencoder the model fails to converge. We did not include these results due to space constraints, but we will add them to the appendix.
>
> Responding to Weakness 4: As is standard in GAN models, no real data is used to synthesize the synthetic data. As described on line 220, to generate synthetic data a random vector of noise $z$ is fed through the embedded generator and then the recovery network to produce the synthetic traces (see Figure 4 and the vertical path that goes from $z$ through Generator\_e and Recovery networks to $\hat{x}$).
>
> Responding to Weakness 5: Yes, this is true. We believe the novelty of our architecture comes from the overall GlucoSynth framework and the use of motif causality, with the integration of prior DP methods a plus that allows for training of our model in a differentially-private manner.
>
> Responding to Question 1: The different colors do not have any real meaning, they were just added for aesthetic purposes. Figure 1 plotted different traces in different colors and Figure 2 shows bars at different times of the day in different colors (e.g., morning has warmer colors and night has darker colors).
>
> Responding to Question 2: Thanks for the opportunity to clarify this. There is no relationship amongst the motifs; motif 1 and 2 are randomly selected glucose motifs (actual motifs shown in Figure 1) and temporal motif 1 and 2 are randomly selected motifs from a cardiology dataset. In each radial graph, there are 24 radial bars from 00:00 to 23:00 for each hour of the day, and the bar value is the percentage of total motif occurrences at that hour across the entire dataset (i.e., value of 10 would indicate that 10\% of the time that motif occurs during that hour in the dataset). The point of this figure is to illustrate that the glucose motifs are not temporally dependent (they show up pretty evenly across the day) and the temporal motifs *are* (they show up only at specific times of the day). For example, glucose motif 2 shows up at every hour of the day whereas temporal motif 2 shows up most frequently in the morning (around 08:00) and evening (around 18:00).
>
> Responding to Question 3: Thanks for this constructive comment. We will expand the figure captions.
>
> Responding to Question 4: Using the weights from $M$, we compute the causality distribution for the sequence of motifs in each trace in $\hat{x}$.
>
> Responding to Question 5: MSE loss on 217 is the error between the original data $x$ and the reconstructed data $\tilde{x}$, the data that was fed through the embedder and then the recovery network. The point of this MSE is to give the autoencoder a way to determine how well it is doing (e.g., how close is the reconstructed trace to the original trace.) A perfect autoencoder perfectly reconstructs the input data. MSE loss on 225 is the error between the batch of real embedded data $x_{et}$ and the batch of synthetically generated data $\hat{x}_{et}$ at the next timestep. The point of this loss is to give the generator a way to compare the temporal distributions it has generated with the real data. Basically, this is how the generator learns to conserve temporal relationships.

---

> > ### Comment · Reviewer_2Sh2 · 2023-08-11
> >
> > I have read through the authors' rebuttal, and I am giving my response accordingly:
> >
> > To Weakness 2: Thanks for your confirmation. This problem is OK for research, but I suspect it will limit its application in reality.
> >
> > To Weakness 3(a): so $m$ is not an arbitrary number. It is the number of motifs in each data partition. Then I wonder if $m$ is available before the training, and how $m$ would affect the generation.
> >
> > To Weakness 3(b): Thanks for the clarification. This is an important detail to me and should not be ignored.
> >
> > To Weakness 4: Thanks for your clarification. But this is confusing to me. Both training and generation of the generator in a standard GAN do not require access to real data. Why is $x_e$ fed into the Generator_e in Fig. 4? How is it used for the Generator_e?
> >
> > To Weakness 5: Thanks for the confirmation.
> >
> > For all the responses to my questions, thank the authors for their detailed explanations.

---

> > > ### Author Response · Authors · 2023-08-11
> > >
> > > Thanks for your response.
> > >
> > > To Weakness 3(a): This is correct, and yes m is available before training (since it is basically a preprocessing step of the real data). This is a good question- the most tangible effect of m on the generator is related to the size of the motifs. Short motifs result in short events being conserved while ignoring longer term ones (e.g., events that repeat over 12+ hours), and vice versa.
> > >
> > > To Weakness 4: The main use of x_e in Generator_e is for computing the stepwise loss (see lines 223-226 in the paper.) This allows the generator to learn to produce realistic next step vectors (focuses on small temporal relations in the synthetic time series.) Without this part, the generator conserves sequences of events and distributional qualities of the dataset, but does not properly synthesize step-by-step values (e.g., the produced traces look really noisy from one timestep to another.)

---

> > > > ### Comment · Reviewer_2Sh2 · 2023-08-16
> > > >
> > > > Thanks for the response. Given that all my questions are clarified, I increase my score.

---

### Official Review · Reviewer_VFZB · 2023-07-07

**Soundness:** 3 good
**Presentation:** 3 good
**Contribution:** 3 good
**Rating:** 5
**Confidence:** 4

**Summary:**

The authors introduce a new differential privacy scheme where other, non-related signal recordings are used to anonymise the signal provided  that they have a motif that seems to predict the time signal forwards in time. For this a time dependent vector is used to contain multiple signals at the same time propagating forwards. Simple RN and LSTM are used.

The models are trained using differentially private stochastic gradient, and with an GAN discriminator component in the network
it is ensured  that  resulting synthetic signal belongs to the same distribution as the actual signals.

The quality of the signal is measured with fidelity, breath, and utility estimates. According to these criteria the novel differential privacy preserving synthetic signals beat the current state of the art.

**Strengths:**

The manuscript is well written and concise explanation of the process.

**Weaknesses:**

The he supporting material to me looks like a copy of manuscript. The paper refers to the Annexes? Please correct

In abstract, the statement below is a contradiction as indicates the novel privacy-preserving GAN framework does not exist, even the manuscript just presents it.  ( it is using GANs and it exists)
"Existing methods fo  time series data synthesis, such as those using Generative Adversarial Networks (GANs), are not able to capture the innate characteristics of glucose data and cannot provide any formal privacy guarantees without severely degrading the utility of the
synthetic data. In this paper we present GlucoSynth, a novel privacy-preserving GAN framework to generate synthetic glucose traces."

I have a problem in understanding the evaluation of the performance. See the question I have in the question part. Please clarify.








**Questions:**

Annex?
I do not understand fully the role of the statistical tests in the evaluation. For fidelity, I would expect that it means that features that are needed to analyse the time trail are preserved, so that one can trust the diagnosis. As the system is generating the signal partly for other trails via the causal motifs, the generated signal may miss something essential, or it may have added components from the other trails. Checking with GAN that the signal is not easily distinguishable from the real traces is a good step and achievement, but at the same time one has to ensure that diagnoses from the original and anonymised trails remain the same for all cases individually - and report the error rate resulting from the anonymisation.


**Limitations:**

Could the method give different diagnostic results after anonymisation. This may be the no-go for the anonymisation. It should be clearly stated that the system is not designed to be used for the diagnostic purposes, but to publish a data set that can be used to train AI solutions - with an extra annotation made for large enough portion of the synthetic examples.

---

> ### Author Rebuttal · Authors · 2023-08-04
>
> We thank the reviewer for their time and review. Responding to the Weaknesses: Thanks for catching these corrections. We will correct the supplemental material submission and adjust the language in the abstract to read: "Previous methods for time series data synthesis ... " We have also responded to the question about the evaluation below.
>
> Responding to the Questions: We thank the reviewer for the opportunity to clarify this important part - the reviewer is correct in their statements, and we evaluated each of these aspects in the 3 parts of the evaluation. First, we used fidelity as a way to verify that on a population level, characteristics of the real data were conserved, i.e., did not change drastically between the real and the synthetic data. To do this, we tested if there was a statistically significant difference in clinically-relevant metrics between the synthetic and real data (if there was not, this is good). Secondly, we used the breadth evaluation to check about what parts of the traces the synthetic data missed, and what anomalous motifs it may have added to the signals (e.g., what components from other trails may have been added), as the reviewer described. Finally, we used the utility part to verify that synthetic traces could be used in place of the real ones (e.g., to ensure that diagnoses from the original and synthetic data were the same) by finding the error in a glucose forecasting task trained on the synthetic data and tested on the real data.
>
> Responding to the Limitations: This is a good point. It is not the authors intention that this synthetic data would be used for individual diagnostic purposes (and note that it would not need to be since there is not a privacy concern with an individual patient sharing their data to their doctor).

---

> > ### Comment · Reviewer_VFZB · 2023-08-17
> >
> > Thank you for the good clarifications. Indeed, the intended use case was my major worry.  This leads to a follow-up question about the actual use case of the DP data sets that can be generated, if not for individual diagnostics.
> >
> > An excellent use case for the generated traces would be to train a classifier on those - presumably not breaking the privacy budget.  After training one should test the inference with human annotated test set to judge the quality of the synthetic data set. A positive finding This would increase my score on the paper.

---

> > > ### Author Response · Authors · 2023-08-17
> > >
> > > Thanks for the follow-up.
> > >
> > > We think a good use case for this is to support development of various ML/AI population based tools (e.g., training other population based models such as glucose forecasting or pattern-based diagnostic models.)
> > >
> > > Thanks for the suggestion. Because the synthetic data is time series traces, it is hard for humans to accurately analyze the quality of the synthetic traces vs. the real ones on their own. Therefore we are not sure a human-based analysis would be the most helpful. However, we do something similar to your suggestion and use our synthetic data to train a glucose forecasting model and then test its results on the real data (see Section 7.3). The models trained on the synthetic data perform well, with their predictions not introducing safety risks (see Appendix E about the Clarke Error Grids). This is one evidence of the utility of our synthetic data for real world tasks.

---

### Official Review · Reviewer_yHP4 · 2023-07-20

**Soundness:** 3 good
**Presentation:** 2 fair
**Contribution:** 3 good
**Rating:** 5
**Confidence:** 3

**Summary:**

This manuscript presents GlucoSynth to generate synthetic glucose traces. The main idea is to conserve relationships among motifs within the time series and to preserve the privacy of the glucose traces. Specifically, the manuscript proposes a motif causality block to learn the causal relationships amongst the motifs and incorporates differential privacy into the framework. The experiments on the clinical dataset demonstrate that the proposed method outperforms the baselines in terms of fidelity, breadth and utility.

**Strengths:**

1. The topic of this manuscript is important. Synthesizing data for sharing is meaningful for collaboration with healthcare institutions.
2. The challenge of generating synthetic glucose traces identified in this manuscript is interesting, i.e., the manuscript claims that " they are more event-driven than many other types of time series."
3. The manuscript designs several evaluation metrics to compare GlucoSynth and previous studies.

**Weaknesses:**

1. The distribution depicted in Figure 2 is not able to support the motivation (glucose traces are event-driven) of this work. It would be better to present the distributions of glucose traces under a subtype of patients, if the distributions are very diverse within the whole population.
2. The contributions of the method are two-folds, i.e., motif causality block and DP. It's not clear how each module contribute to the whole model through the experiments. A well-designed ablation study would be better.
3. The organization of the manuscript should be improved. Several important results are in Appendix, while the method is too long.

**Questions:**

1. Can the temporal dependencies learned by the model be visualized or aligned to medical knowledge? Are they approved by healthcare experts, e.g., doctors?
2. As descirbed the Sec. 7.3, an LSTM is trained for glucose forecasting. How many times does the model run? Have you tested with other models? Are there any differences between LSTM and other models?
3. What about the scalabilty of the proposed model? For instance, do you test the performance of GlucoSynth on a small dataset? How much samples are required at least for training GlucoSynth usable? Have you evaluate the models on other datasets, as mentioned in Limitations?

**Limitations:**

1. The authors can adequately recognize the limitations of this work. It is worth emphasizing that the motivation is not supported well by the statistics and the experiments. Besides, the scalabilty of the model is not explore.
2. There is no significant potential negative societal impact of this work.

---

> ### Author Rebuttal · Authors · 2023-08-04
>
> We thank the reviewer for their time and review. Responding to Weakness 1: The main point of Figure 2 was to show that glucose motifs are not temporally-dependent, and provide insight about why methods that only focus on conserving temporal relationships fail. The motivation of the work was to develop a method that produces high-quality synthetic glucose traces. As a way to explain perhaps why other methods fail to produce high-quality synthetic traces, we describe glucose characteristics and how they are driven by behaviors (a clinically-supported fact) and not temporally dependent in Section 3.2 using Figure 2.
>
> Responding to Weakness 2: We thank the reviewer for this suggestion. Due to space constraints we decided not to include an ablation study and instead focused on comparing and evaluating the overall framework performance. That being said, this would be a nice addition for the Appendix.
>
> Responding to Weakness 3: We thank the reviewer for this suggestion. Since we have an unusual architecture, we wanted to make sure enough details were provided about the methods, and due to space constraints, felt that putting some of the results in the appendix was a fair trade-off.
>
> Responding to Question 1: Yes. We plotted motif behaviors on graphs and worked with a group of diabetes clinicians during model development to ensure our behaviors were human physiologically possible and would not cause patient harm if used for medical tasks.  As a way to quantitatively test this, we plotted Clarke Error Grids in the Utility experiments (see Appendix E), which is a basis for evaluating the safety of diabetes-related medical devices.
>
> Responding to Question 2: Thanks for these interesting questions. We run the experiment 10 times and train the LSTM for 10,000 epochs. We have tested with other models including RNNs, attention-based models and other LSTM architectures (such as bidirectional LSTMs). We show the results for this particular glucose-focused LSTM architecture because it performs the best at the glucose forecasting task compared to the other architectures. That being said, even when using the other models we still find that the RMSE decreases as our privacy budget increases, and that GlucoSynth performs the best compared to previous methods. We will add these additional details to Appendix E.
>
> Responding to Question 3: Thanks for this important question. We have tested GlucoSynth on smaller datasets and, as is a common trade-off with differential privacy methods and generative models, find empirically that utility degrades as population sizes decrease. We are currently experimenting with population sizes vs. privacy trade-offs (e.g., what is the minimum number of traces needed to provide decent privacy guarantees). In addition, we are working on optimizing the model to make it more scalable.

---

> > ### Comment · Reviewer_yHP4 · 2023-08-14
> > **Thanks for the response**
> >
> > I have read the rebuttal. I still have some concerns:
> >
> > 1. To Weakness 1: (c) and (d) are from a cardiology dataset, why they can be used to support the motivation of this paper?
> >
> > 2. To Weakness 2 and Question 3: do you have some preliminary results or observations?
> >
> > Thanks again for the clarification.

---

> > > ### Author Response · Authors · 2023-08-17
> > >
> > > Thanks for the follow-up.
> > >
> > > 1. Right, so the point of the cardiology dataset was to have a different dataset that is temporally dependent that we could compare the glucose motifs with, to show what we mean by the glucose motifs not being temporally dependent. (You could pick any temporal dataset and substitute in results here, we just chose a cardiology one for this.)
> > >
> > > 2. Based on preliminary results we find in general that utility degrades as population sizes decrease. Population sizes of >~10,000 patients have good privacy trade-offs, but any smaller and we begin to lose utility (particularly do worse on breadth results.)

---

> > > > ### Comment · Reviewer_yHP4 · 2023-08-18
> > > > **Thanks for the response.**
> > > >
> > > > I still have some concerns on this paper. The glucose dynamics is not convincing for me. It would be better to explain the details of the data collection and motif discovery. Moreover, the evidence from medical domains and experts is also needed.
> > > >
> > > > The overall idea is interesting and the solution is novel. I would like to increase my rating.

---

### Official Review · Reviewer_Q4qV · 2023-07-23

**Soundness:** 3 good
**Presentation:** 2 fair
**Contribution:** 3 good
**Rating:** 5
**Confidence:** 3

**Summary:**

In this paper, the problem of generating differentially private synthetic glucose traces using a GAN framework, by conserving the relationships among glucose events or motifs within the traces, as well as the temporal dynamics.

**Strengths:**

This paper presents GlucoSynth, an approach that generates private and high-utility synthetic data. The method combines some existing ideas in a clever way. The work is interesting and the results show an improvement to the previous literature on the topic.

**Weaknesses:**

- The major issue is that more information is needed for proving the privacy guarantees offered by this method. In section 6, the addition of differential privacy is discussed, however, the explanation is brief and more rigorous proofs are needed.

- Moreover, some more details are needed when explaining the motif causality block in section 4.2. For instance, when discussing $\tau$ as a hyperparameter, then stating that "we use $\tau = 48$, corresponding to 4 hours of time", is this related to the experiments or why is this used here?

**Questions:**

- More detailed privacy proofs needed.
- In section 4.2, it is stated that $\tau = 48$ is used, corresponding to 4 hours of time. Is this a common $\tau$ value or does this just work for the experiments in this paper.

**Limitations:**

The authors discuss the limitations of the work.

---

> ### Author Rebuttal · Authors · 2023-08-04
>
> We thank the reviewer for their time and review. For our privacy integration, we use a straightforward application of prior differential privacy methods to our framework. Since we have not developed any new differential privacy mechanism, and are using established privacy techniques in a standard way, we do not believe a formal proof is required.
>
> Related to the comments about needing more details for the motif causality block, we provide additional justification for the hyperparameter selection (including for $\tau$) in the Appendix B. $\tau = 48$ was chosen because it is a clinically significant threshold: the effect of any behaviors on glucose occur within 4 hours of the event (e.g., the effect from eating a meal – a rise in glucose – will occur within 4 hours after eating.) If there are other questions about the motif causality block, we are happy to answer them.

---

> > ### Comment · Reviewer_Q4qV · 2023-08-14
> >
> > I thank the authors for their clarifications. About the privacy, although I do agree that it is straightforward, it is good to explain it better.
> > Based on the reviews and the discussions, I raise my score to a 5.

---

### Decision · Program_Chairs · 2023-09-21

**Decision:**

Accept (poster)

**Comment:**

The paper presents a method for generation of private synthetic time course data using an innovative model that takes into account prior knowledge of the physical constraints underlying the data. All reviewers recommend acceptance as the authors were able to answer their questions, although most consider this only borderline accept.

I am also in favour of acceptance, as I like the radically new kind of approach for private synthetic data generation that may prove useful in other applications too.